# A chemical interpretation of protein electron density maps in the worldwide protein data bank

Sen Yao[1,2,3], Hunter N. B. Moseley 🔘 [1,2,3,4,5]*

1 Department of Molecular & Cellular Biochemistry, University of Kentucky, Lexington, Kentucky, United States of America, 2 Markey Cancer Center, University of Kentucky, Lexington, Kentucky, United States of America, 3 Resource Center for Stable Isotope Resolved Metabolomics, University of Kentucky, Lexington, Kentucky, United States of America, 4 Institute for Biomedical Informatics, University of Kentucky, Lexington, Kentucky, United States of America, 5 Center for Clinical and Translational Science, University of Kentucky, Lexington, Kentucky, United States of America

* hunter.moseley@uky.edu

## Abstract

High-quality three-dimensional structural data is of great value for the functional interpretation of biomacromolecules, especially proteins; however, structural quality varies greatly across the entries in the worldwide Protein Data Bank (wwPDB). Since 2008, the wwPDB has required the inclusion of structure factors with the deposition of x-ray crystallographic structures to support the independent evaluation of structures with respect to the underlying experimental data used to derive those structures. However, interpreting the discrepancies between the structural model and its underlying electron density data is difficult, since derived sigma-scaled electron density maps use arbitrary electron density units which are inconsistent between maps from different wwPDB entries. Therefore, we have developed a method that converts electron density values from sigma-scaled electron density maps into units of electrons. With this conversion, we have developed new methods that can evaluate specific regions of an x-ray crystallographic structure with respect to a physicochemical interpretation of its corresponding electron density map. We have systematically compared all deposited x-ray crystallographic protein models in the wwPDB with their underlying electron density maps, if available, and characterized the electron density in terms of expected numbers of electrons based on the structural model. The methods generated coherent evaluation metrics throughout all PDB entries with associated electron density data, which are consistent with visualization software that would normally be used for manual quality assessment. To our knowledge, this is the first attempt to derive units of electrons directly from electron density maps without the aid of the underlying structure factors. These new metrics are biochemically-informative and can be extremely useful for filtering out low-quality structural regions from inclusion into systematic analyses that span large numbers of PDB entries. Furthermore, these new metrics will improve the ability of non-crystallographers to evaluate regions of interest within PDB entries, since only the PDB structure and the associated electron density maps are needed. These new methods are available as a well-documented Python package on GitHub and the Python Package Index under a modified Clear BSD open source license.

**Data Availability Statement:** An older version of this manuscript is available on the bioRxiv preprint server: https://doi.org/10.1101/613109. Also the described software is available on GitHub (https://github.com/MoseleyBioinformaticsLab/pdb_eda) and the Python Package Index (https://pypi.org/

project/pdb-eda/), the documentation is available on ReadTheDocs (https://pdb-eda.readthedocs.io/en/latest/), and all results are available on a FigShare repository (https://doi.org/10.6084/m9.figshare.7994294).

**Funding:** HNBM received National Science Foundation (https://www.nsf.gov/) award 1419282. The funders had no role in study design, data collection and analysis, decision to publish, or preparation of the manuscript.

**Competing interests:** The authors have declared that no competing interests exist.

## Introduction

Proteins are active components in the biochemical implementation of biological processes, and understanding their structure is important for interpreting their biochemical functions. The Worldwide Protein Data Bank (wwPDB, www.wwpdb.org) [1] is the international organization that manages the Protein Data Bank (PDB, www.rcsb.org) [2], the central repository of biological macromolecules structures. Thousands of structures are deposited into the wwPDB every year, but their data quality can vary significantly from structure to structure, and even region to region within a structure. Low-quality data can cause problems for both a single macromolecule structure inspection and aggregated systematic analyses across hundreds or thousands of structural entries [3, 4]. Thus, the analysis and interpretation issues caused by the presence of low-quality structural data are pushing the structural biology community to pay more attention to the quality of deposited structural entries [5]. The wwPDB has initiated several efforts to improve the quality of entries being deposited, including launching a deposition, biocuration, and validation tool: OneDep [6]. Many data quality measures are now available for PDB structures, such as a resolution, B-factors, MolProbity clashscores [7], to name a few. However, low-quality regions can still exist even in structures with very good metrics of global structural quality, as shown in the overlay of structures with electron density maps in Fig 1. These low-quality regions arise from structural model and electron density mismatches that can be due to a variety of reasons including problems with regional protein mobility that can often lead to an apparent lack of electron density [8–10], data processing [11, 12], or model fitting [10, 13, 14]. These mismatches often occur around bound ligands where a lot of interesting biological activities happen, making the analysis of protein sequence-structure-function

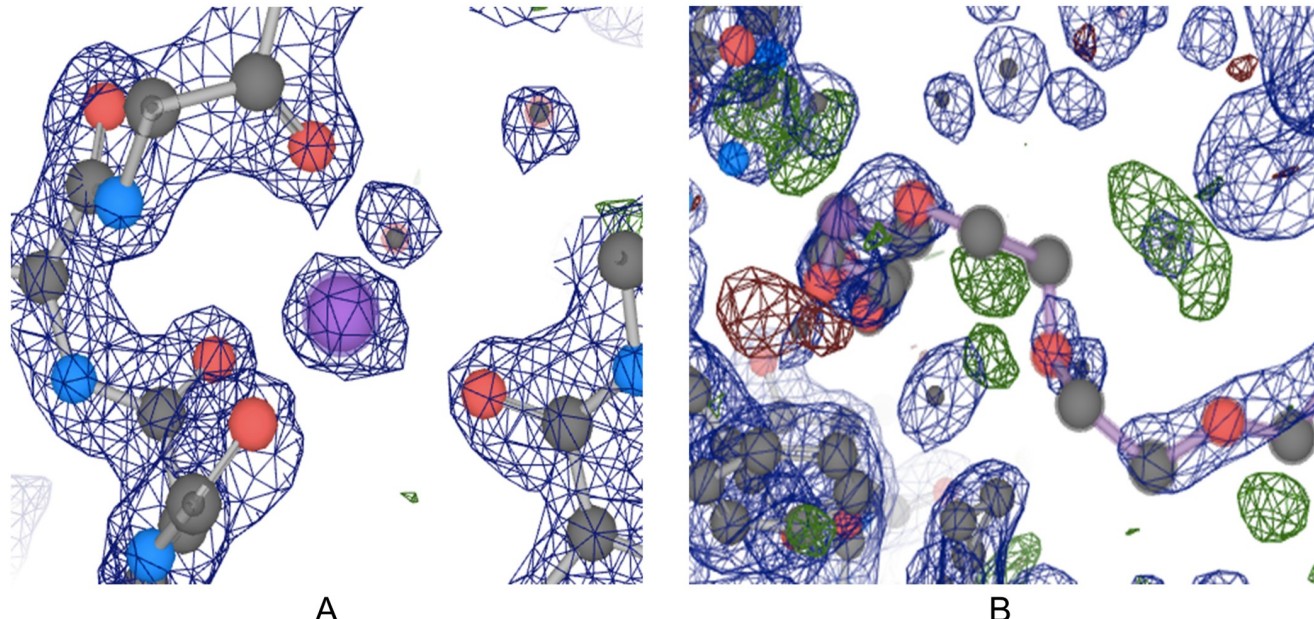

**Fig 1. Model and difference electron density maps.** Both panels A and B are from the same structure, PDB ID: 3B1Q, and are centered around the ligands B.330 (A) and P.33 (B). Blue meshes represent the model electron density. Green and red meshes represent discrepancies between experimental data and the structure model. The structure has very good overall quality, as demonstrated by a resolution of 1.7 Å, an R-factor value of 0.176, and an R-free value of 0.207. The panel A shows a high-quality region in the structure where the experimental data and model match very well. And the panel B shows that there are still low-quality regions within the structure, as demonstrated by the green and red blobs around the coordinating ligand residue ("ligand" refers to the coordination chemistry definition of this word).

relationships more difficult. Therefore, the evaluation of structure quality, especially around regions of interest, is paramount before accurate structural inferences can be made.

Driving improved evaluations of structure quality are newer deposition requirements like mandatory deposition of structure factors (x-ray structures) and constraints (NMR structures) starting from 2008 [15], NMR-assigned chemical shifts from 2010, and 3DEM volume maps from 2016 [7]. The inclusion of underlying experimental data used in structure determination enables researchers to better validate structural models, improving the inferences they can make from these structures. For x-ray crystallographic structures, electron density maps enable a direct comparison between the observed electron density Fo to calculated electron density Fc based on the structural model. The 2Fo-Fc map represents the electron densities surrounding well-determined atoms in the model across a three-dimensional (3D) space and the Fo-Fc map represents the electron density discrepancies between the observed and calculated electron density across a 3D space. For x-ray crystallographic PDB entries with deposited experimental data, sigma-scaled electron density maps are made available by the PDB in Europe (PDBe) [16]. Previously from about 1998 to 2018 [17], electron density maps were made available by the Uppsala Electron Density Server (Uppsala EDS), which was created and maintained outside of the PDB [18]. However, the PDBe uses newer methods to provides higher quality density maps, which prompted the retirement of the Uppsala EDS by 2018. Many electron density map viewers [19–22] exist for manually examining the quality of a model versus its electron density; however, this software and evaluation approach is not suitable for batch analysis of hundreds of structures. Also, these sigma-scaled electron density maps are in arbitrary units of electron density, with no direct physicochemical meaning. This normally does not affect the visualization of electron densities and is a by-product of creating maps with a summative intensity of zero (zero-sum) across the whole map, which is done primarily for visual simplification during modeling [23, 24]. But this zero-sum representation can be detrimental for understanding a model, especially a local region of a model, where the number of electrons represented by the density or density discrepancy would be useful for evaluation. While methods exist that can derive electron density maps on an absolute scale, these methods require a reanalysis of the underlying structure factors with software packages that are not designed for automated use across large numbers of structural entries [25, 26]. Due to these limitations, we have developed a new method that derives a conversion factor from the arbitrary electron density units of a given electron density map with corresponding PDB entry into the absolute value of electrons per angstroms cubed, without the need to reprocess and reanalyze the underlying structure factors. With this conversion factor, we have developed new evaluation methods that normalize electron density and electron density discrepancies into estimated quantities of electrons. These new electron discrepancy values can provide chemically-informative information for evaluating structural models or for filtering structure entry regions for inclusion into systematic analyses that span large numbers of PDB entries.

## Methods

### Calculating the electron density ratio for atoms, residues, and chains

A workflow of the analysis is shown in Fig 2. Structural data (in PDB format) was downloaded from wwPDB on Jul 3, 2018, and their electron density data (in CCP4 format), if available, was acquired from the PDBe website [16]. Structural data was processed using a self-developed parser and Biopython [27]. Electron density data was analyzed according to the CCP4 suite [28] format guidance. The electron density map is represented as a 3D array in the data, which corresponds to voxels in the real space. An electron density voxel with a density value greater than $1.5\sigma$ of all voxels is considered significant for 2Fo-Fc maps, and $3\sigma$ for Fo-Fc maps. For

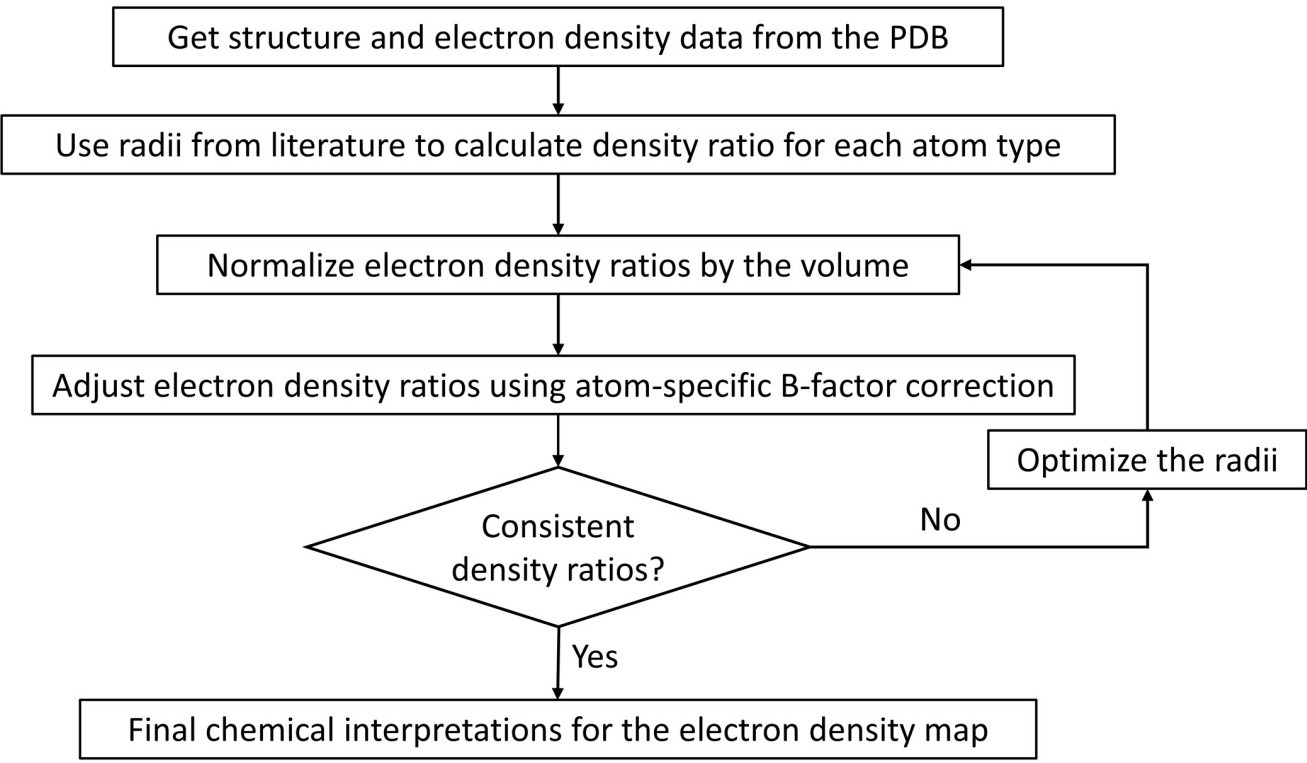

**Fig 2. Workflow of the electron density analysis.**

structures with electron density data available, symmetry operations were performed to include the surrounding environment for the modeled structure.

To calculate the total electron density around each atom, we initially used the radii from literature [29] and calculated the sum of all densities within the corresponding radius. The voxel center is used when calculating the distance from a density voxel to an atom. Different atoms (without hydrogens) from the 20 common amino acids are categorized into 13 atom types as shown in S1 Table. Electron density ratio ($r_i$) is defined as the total density of all associated voxels ($\rho_m$) divided by the number of electrons ($Z_i$) for a given atom $i$,

$$r_i = \frac{\sum \rho_m}{z_i} \tag{1}$$

As the unit for electron density is $e\text{Å}^{-3}$, the electron density ratio thus has the unit as $\text{Å}^{-3}$. The total density is adjusted by a factor of the occupancy of the atom. Since hydrogen is normally not resolvable within electron density maps, their electrons were added to their bonded atom. A table of electron counts used for each atom is shown in S2 Table.

After all atom electron densities are calculated, they are aggregated into residue and chain densities where the residue cloud contains at least 4 atoms and the chain cloud contains at least 50 atoms. The overlapping density voxels between two or more atoms are only counted once through the aggregation. The total number of electrons is calculated by adding contributing atom's electron numbers together. Residue ($r_r$) and chain ($r_c$) density ratios are then calculated accordingly.

## Normalizing the electron density ratio by the number of voxels

To smooth the representation of continuous electron densities using discrete voxels, the electron density ratio is then normalized by the median volume (in number of voxels) of a given atom type. If we denote the original density ratio as $r_i$ and the volume of a given atom $i$ with atom type $t$ as $V_i$, and the median volume of all atoms with atom type $t$ as $median(V_t)$, the normalized density ratio $r_{i-norm}$ can be defined as follow:

$$r_{i-norm} = r_i * median(V_t)/V_i \tag{2}$$

## Correcting the unit electron density by the atom B-factor

As the actual value of the density ratio is highly specific to individual structures, we then define a more universal measure as the chain deviation fraction ($f_i$) for a given atom $i$ as:

$$f_i = (r_{i-norm} - median(r_c))/median(r_c) \tag{3}$$

The dispersion of electron density around an atom can be approximated using the B-factor of the given atom. The chain deviation fraction and logarithmic B-factor have a linear correlation both statistically and visually, and thus a slope ($s_t$) of chain deviation fraction over logarithmic B-factor can be calculated for each atom type and for every structure. If there are less than three points for a given atom type, the median slope over 1000 random structures is used. Then for each individual atom $i$ with atom type $t$, its unit electron density can be corrected by its deviation from the median B-factor:

$$f_{i-corrected} = f_i + (log(b_i) - median(log(b_t))) * s_t \tag{4}$$

$$r_{i-corrected} = f_{i-corrected} * median(r_c) + median(r_c) \tag{5}$$

## Optimization of radii

After the initial calculation, the median density ratios of different atom types were still quite different from each other. Thus, to achieve a more uniformly interpretable density ratio within a structure as well as across structures, an optimization of radii was performed. First, we tested the radius for each atom type on 100 random structures and obtained an initial estimation of the radii. The metric we used to optimize was the median of corrected chain deviation fraction ($f_{i-corrected}$) for a given atom type. Based on the results from the initial step, we then optimized one atom type at a time on 1000 randomly selected structures. For every iteration, the atom type that has the largest deviation from last round was optimized. Different radii were tested for the given atom type and the radius that has a median corrected chain deviation fraction closest to zero was picked out. At the end of each optimization, the set of B-factor slopes were updated as well. This process continued until the median chain deviation fraction for all atom types were smaller than 0.05. The final set of radii were then tested on another 1000 random structures and the whole PDB database for validation.

## Adding an $F_{000}$ term

The average value of the 2mFo-DFc map (i.e. a Sigma-A weighted map) is practically zero for most of the structures in the PDB. Theoretically, an $F_{000}$ term should be added to get the proper number of electrons on an absolute scale. Unfortunately, not all structure factor

programs provide the $F_{000}$ value. So as an estimation, we add up the numbers of electrons for all the atoms of a model in the unit cell, including symmetry structure units and modeled water molecules,

$$F_{000} = \sum n_t Z_t \tag{6}$$

Where $n_t$ is number of atoms of element t in the asymmetric unit and $Z_t$ is the number of electrons (atomic number) of element t. This estimated $F_{000}$ term is then divided by the unit cell volume V in $Å^3$ and added to the density values of all the voxels.

### Design, implementation, and distribution of the above methods

These new methods are implemented in a Python package, pdb-eda. It is written in major version 3 of the Python program language and is available on GitHub, https://github.com/MoseleyBioinformaticsLab/pdb_eda, and the Python Package Index (PyPI), https://pypi.org/project/pdb-eda/. There are three main parts of pdb-eda: the pdb parser, the ccp4 parser, and the electron density analysis. Starting from a PDB id, pdb-eda can either read a local pdb or ccp4 file or download it on the fly. Intermediate and final results of all three parts can be accessed via either importing as a library or using the command line interface. Many options are available for handling and processing the data with the details documented in the package guide and tutorial files. As part of the development process, new versions are updated on GitHub regularly. The version that is described and implemented for this paper has been frozen, tarballed, and published on FigShare (https://doi.org/10.6084/m9.figshare.7994294), along with all the result files and codes in generating all results, figures, and tables.

## Results

We downloaded and used a total of 141,763 wwPDB entries, of which 106,321 structures have electron density maps available and suitable for the analysis in this study. The assumption of this study is based on a fundamental rule for electron density construction, that is the electron density is proportional to the number of electrons. However, after being deposited into the PDB, this information of absolute value of electrons is hard to derive and is inconsistent across structures. Different structures can vary a lot in terms of density ratios, due to the quality of the crystal or the choice of data processing software. Therefore, we need an internal measure to enable a consistent interpretation within and across structures. If we simply use the radii from the literature and do not apply any correction, the median of atom density ratio shows that the density ratios are inconsistent within a single structure for atoms, residues, and chains, as illustrated in Fig 3 Panels A-C as Sina plots, an enhanced jitter strip chart that visualizes the distribution of each dataset for better visual comparison [30]. The atom density ratios span over the largest range, while the chain density ratio has the smallest range. Therefore, we chose the chain deviation fraction as a reliable measure to optimize all atom types to the same level.

### Volume and B-factor adjustment

An example of the volume distribution for a single structure is shown in Fig 4. Ideally, atoms of the same atom type should occupy the same volume. However in reality, different parts of a structure may be more or less ordered within the crystal than others. Also, during the reconstruction of the electron density data from complex structure factors, continuous electron density through space is represented as a point density value for every voxel. And the use of the point density to estimate the whole voxel depends on the smoothness of the density function. Moreover, based on the placement of the atom in relation to the voxel and the selection of grid

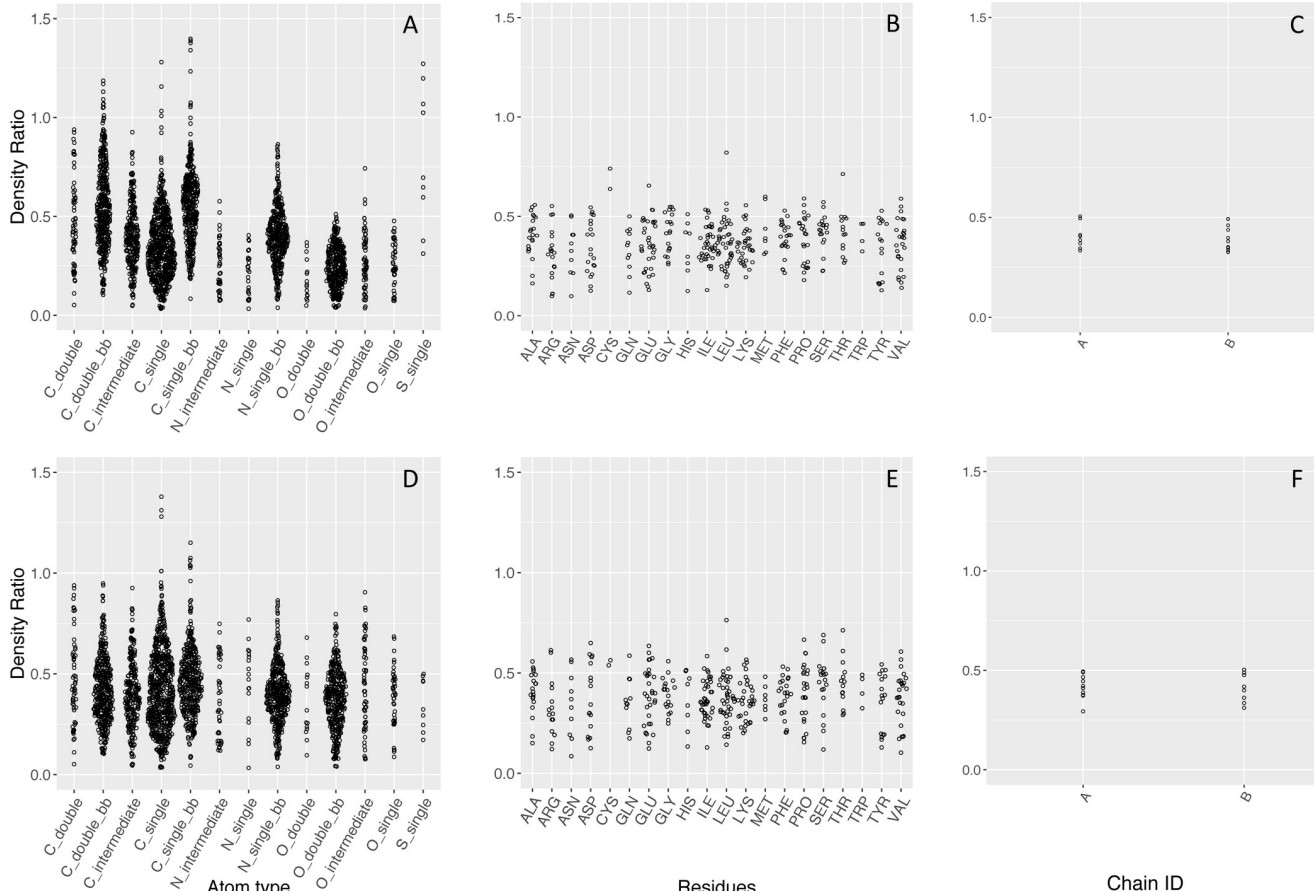

**Fig 3. Sina plots of density ratio for atoms, residues, and chains, before (Panels A-C) and after (Panels D-F) radii optimization.** PDB ID: 3UBK. The atom density ratios have the largest range, and chain density ratios have the smallest range, which can be used as internal standard to optimize the atom ratios to.

length, the inclusion of a voxel is an all-or-none decision. Thus, we performed the volume normalization to minimize these effects.

B-factor measures the temperature-dependent atomic displacement in a crystal. As a result, it is inversely correlated with the total electron density within a distance of an atom, and thus the density ratio. After examining several different relationships between density ratio and B-factor, both statistically and visually, the chain deviation fraction versus logarithmic B-factor demonstrated the strongest linear correlation, and thus was used for the correction. This is not surprising due to the expected non-linear relationship of B-factors [10]. Fig 5 provides an example of this relationship.

The volume normalization correction helps to reduce the high variability of the density ratio while the B-factor correction improves the symmetry of each atom density ratio distribution. Fig 6 illustrates the distributions of atom density ratios before and after each step (Panels A-C). After both adjustments, the atom density ratio is coherent within each atom type, though it is still uneven between atom types (Panel C).

## The final set of radii after optimization

Fig 6, Panel D illustrates the distributions of atom density ratios after radii optimization, where different atom types have much more similar median ratios. Moreover, the standard

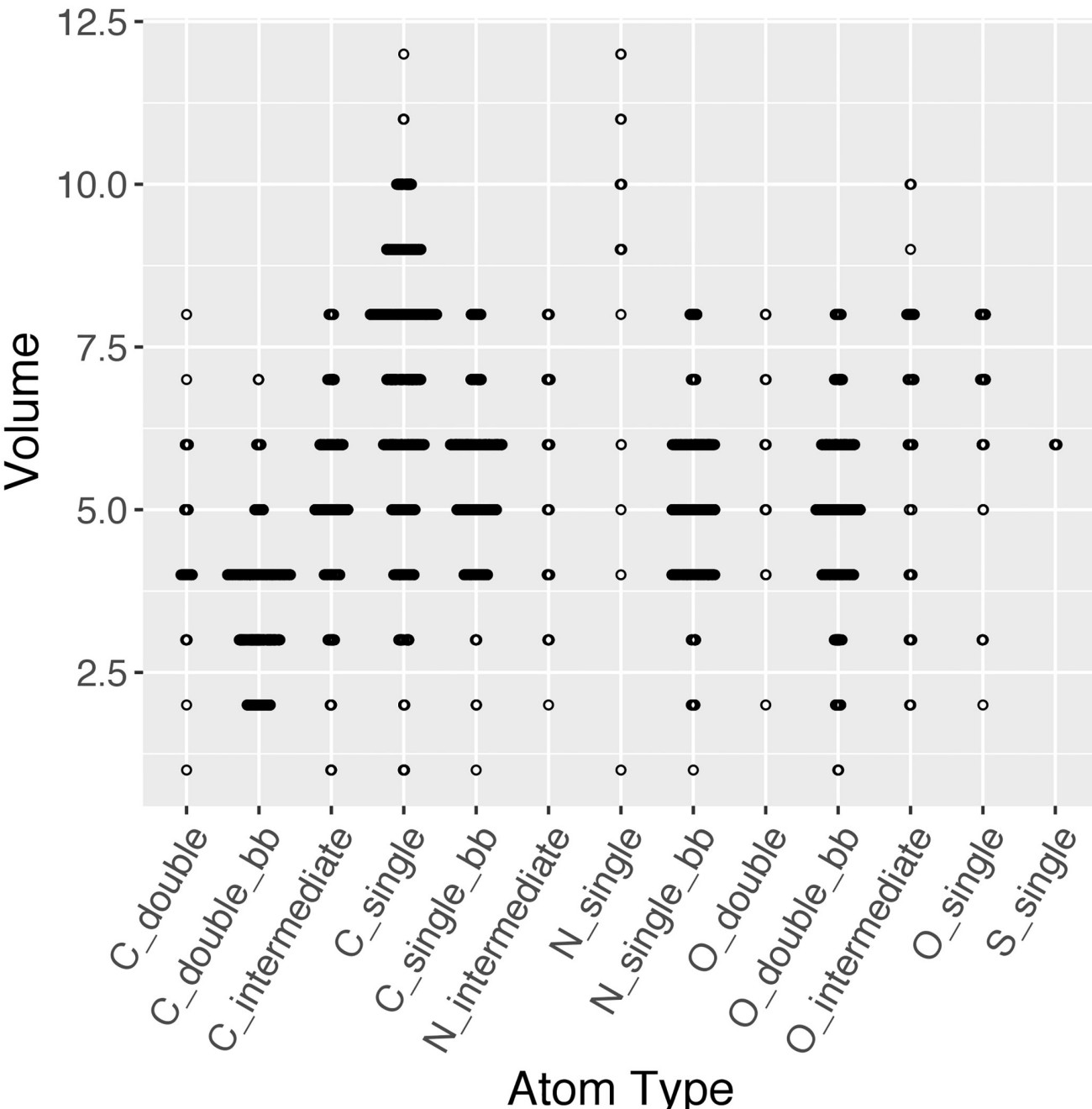

**Fig 4. Sina plot for the volumes of each atom type.** PDB ID: 3UBK.

deviation of the overall distribution drops to 0.09, representing a 40% decrease in the calculated density ratio variability. This improvement percolates through to the residue and chain level, but is not as obvious (see Fig 3, Panels D-F). A comparison of initial and final set of radii is shown in Table 1. In general, most of the backbone atoms decrease in radius, while most of the optimized radii on the side chain are larger than those on the backbone. This is due to the lower order and higher flexibility of the side chain atoms, which practically requires a larger radius to capture the expected number of electrons.

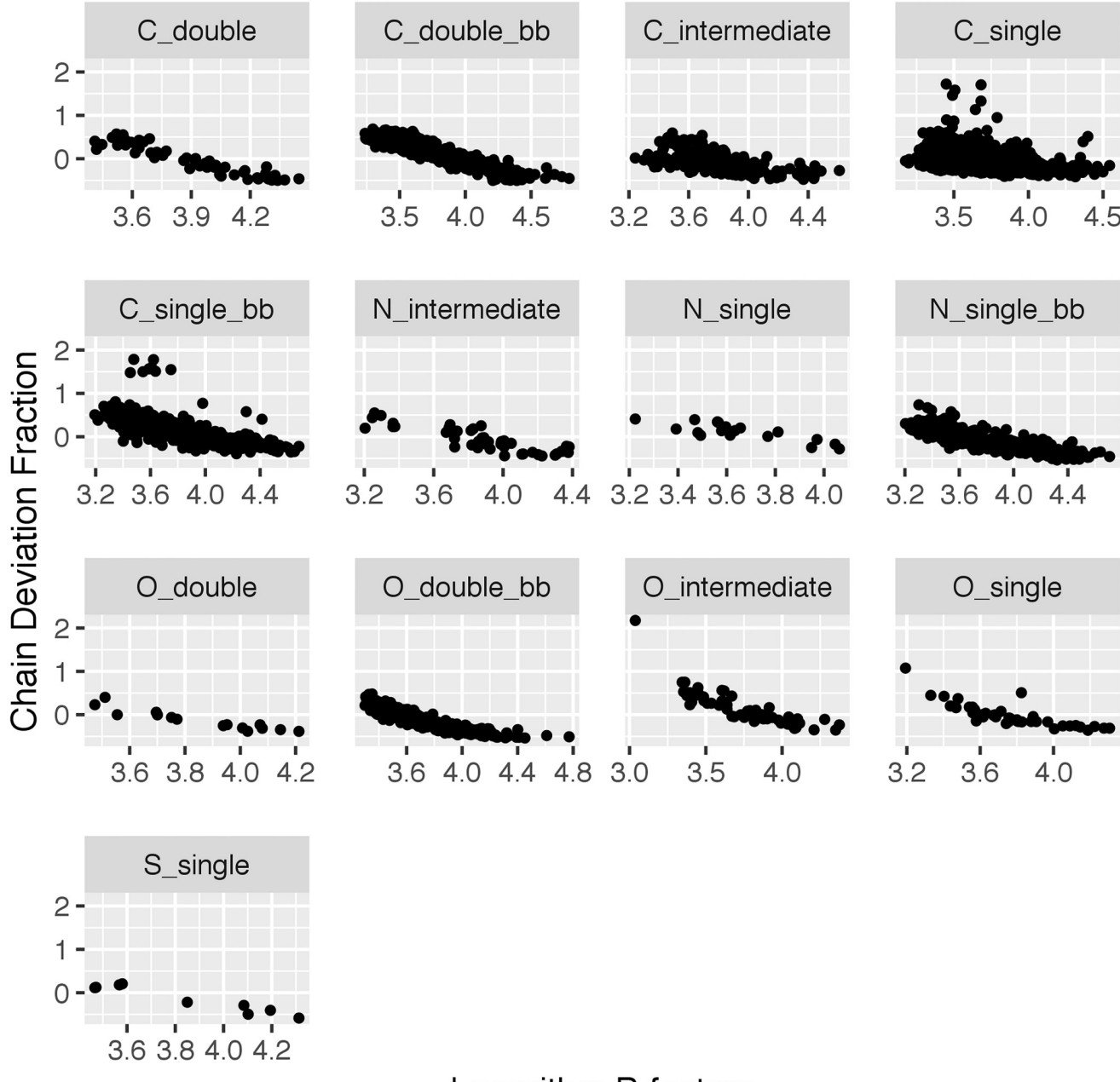

**Fig 5. Correlation between chain deviation fractions and logarithm B-factors for each atom type.** PDB ID: 3UBK.

The radius of sulfur changes the most as compared to the other elements. This could be due to how most software construct electron density data from structure factor and associated phase via Fourier transforms. The electron density is approximated with a Gaussian distribution, and its variance is affected mainly by the B-factor. Thus, the final optimized radius is a combination of the actual radius of the atom, the displacement of an atom center, as well as the thermal motion of the atom (B-factor). Moreover, studying the behavior of sulfur atoms could be useful for other less common elements such as metal ions.

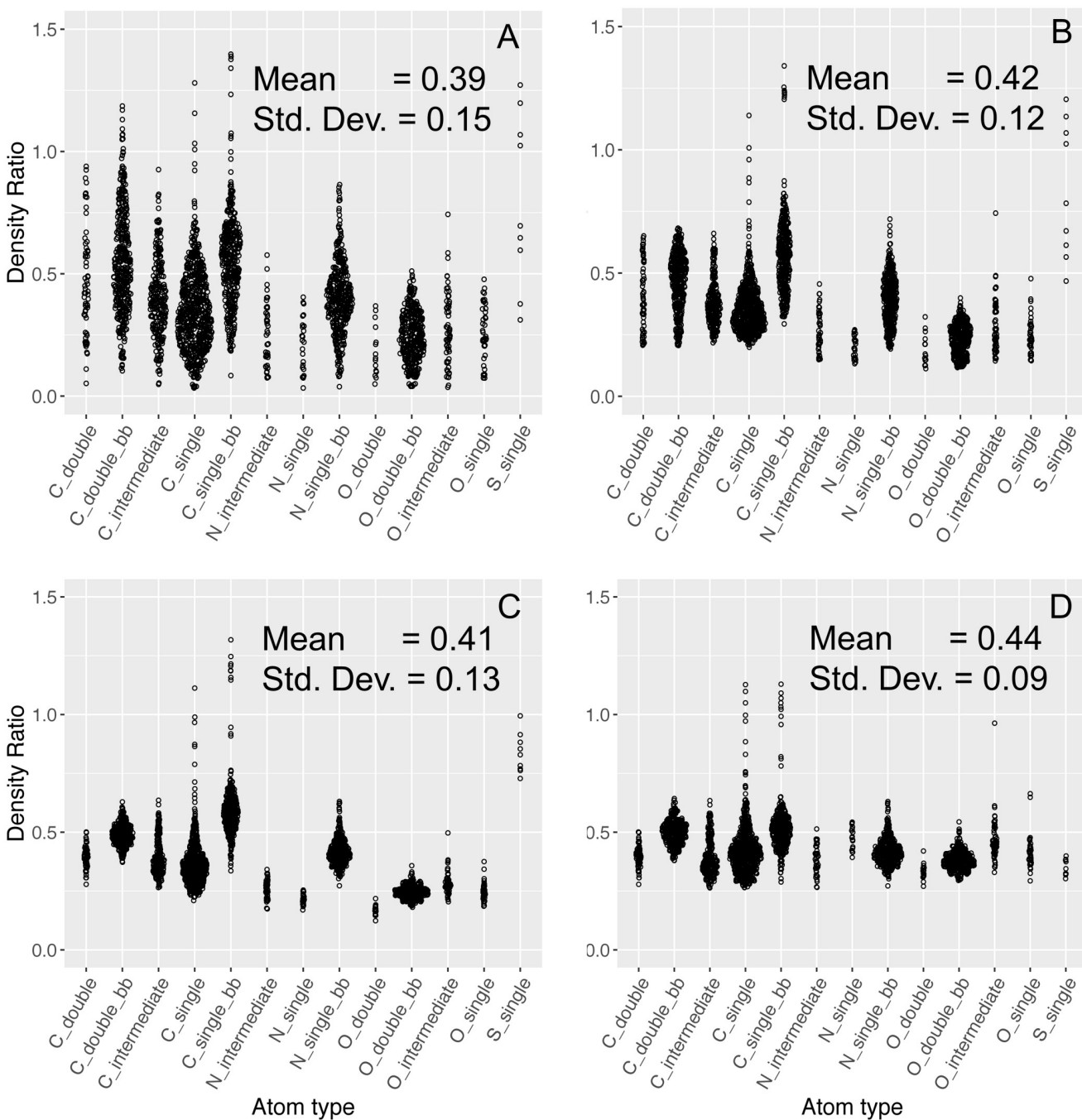

**Fig 6. Atom density ratios at each major step of improvement.** A) Original, B) After volume normalization, C) After B-factor correction, D) After radii optimization. PDB ID: 3UBK. After each major step, the overall distribution of the atom density ratios generally becomes less spread.

## Overview of density ratio for the whole PDB database

The final set of radii was first tested on another 1000 random structures and all atom types held true to have no more than a 5% chain deviation fraction. It was then applied to all PDB structures that had usable electron density data, and the results are shown in Fig 7. For all atom types, the distributions center around 0, which indicates the set of optimized radii yields

**Table 1. The atom radii before and after radii optimization.**

| Atom Type | Original Radius (Å) | Optimized Radius (Å) |
|---|---|---|
| C_single | 0.77 | 0.84 |
| C_single_bb | 0.77 | 0.72 |
| C_double | 0.67 | 0.67 |
| C_double_bb | 0.67 | 0.61 |
| C_intermediate | 0.72 | 0.72 |
| O_single | 0.67 | 0.80 |
| O_double | 0.60 | 0.71 |
| O_double_bb | 0.60 | 0.77 |
| O_intermediate | 0.64 | 0.71 |
| N_single | 0.70 | 0.95 |
| N_single_bb | 0.70 | 0.70 |
| N_intermediate | 0.62 | 0.77 |
| S_single | 1.04 | 0.75 |

"_bb" identifies backbone atom types.

consistent measures of density ratios across structures. As shown in S1 Fig, for high-quality structures with a resolution smaller than 1.5Å, the chain deviation fraction illustrates tighter distributions with modes above 0, because of the narrower electron dispersion around atoms in the experimental data. As the resolution gets worse, this distribution tends to broaden for all atom types with the modes smaller than 0.

## $F_{000}$ term and the absolute scale

Analysis of electron density on an absolute scale (i.e. in units of $e/Å^3$) requires the value of $F_{000}$ and the unit cell volume. As the shape of the density matters more than the absolute scale in the structure modeling, most maps lack this $F_{000}$ term. Therefore, the mean value of the 2mFo-DFc map is practically zero across the whole PDB, as shown in Fig 8. To get the true electron density values, we would need to add an $F_{000}$ term to the set of Fourier coefficients going into the calculation of the map. However, as Fig 9 shows before and after adding the estimated $F_{000}$ term, it makes very little contribution to the overall absolute electron density values. This is probably due to missing the contribution from the bulk solvent, which is not easy to retrieve without extra data and software packages [25, 26]. Furthermore, the bulk solvent is estimated in different ways depending on resolution [31, 32]. Thus, an $F_{000}$ term as theoretically represented in textbooks and papers [33, 34] is not easy to calculate and likely depends on software parameters used in the creation of the map. Therefore, the chain median is used as a conversion factor to relate all electron density values back to the absolute scale.

## Evaluative use-case for the electron density conversion factor

One of the most important applications of this work is to estimate the difference density map in terms of electrons. The total difference in expected vs actual electron density can be represented in electron units by dividing the total electron densities by the conversion factor (the median of chain density ratios). As shown in Fig 10, the Fo-Fc map overlaying the 2Fo-Fc map and structure model show several positive (green) and negative (red) density blobs between the measured density from the experiment and the density explained by the given model. On panel A, most of the discrepancies are below six electrons, which can be reasonably interpreted

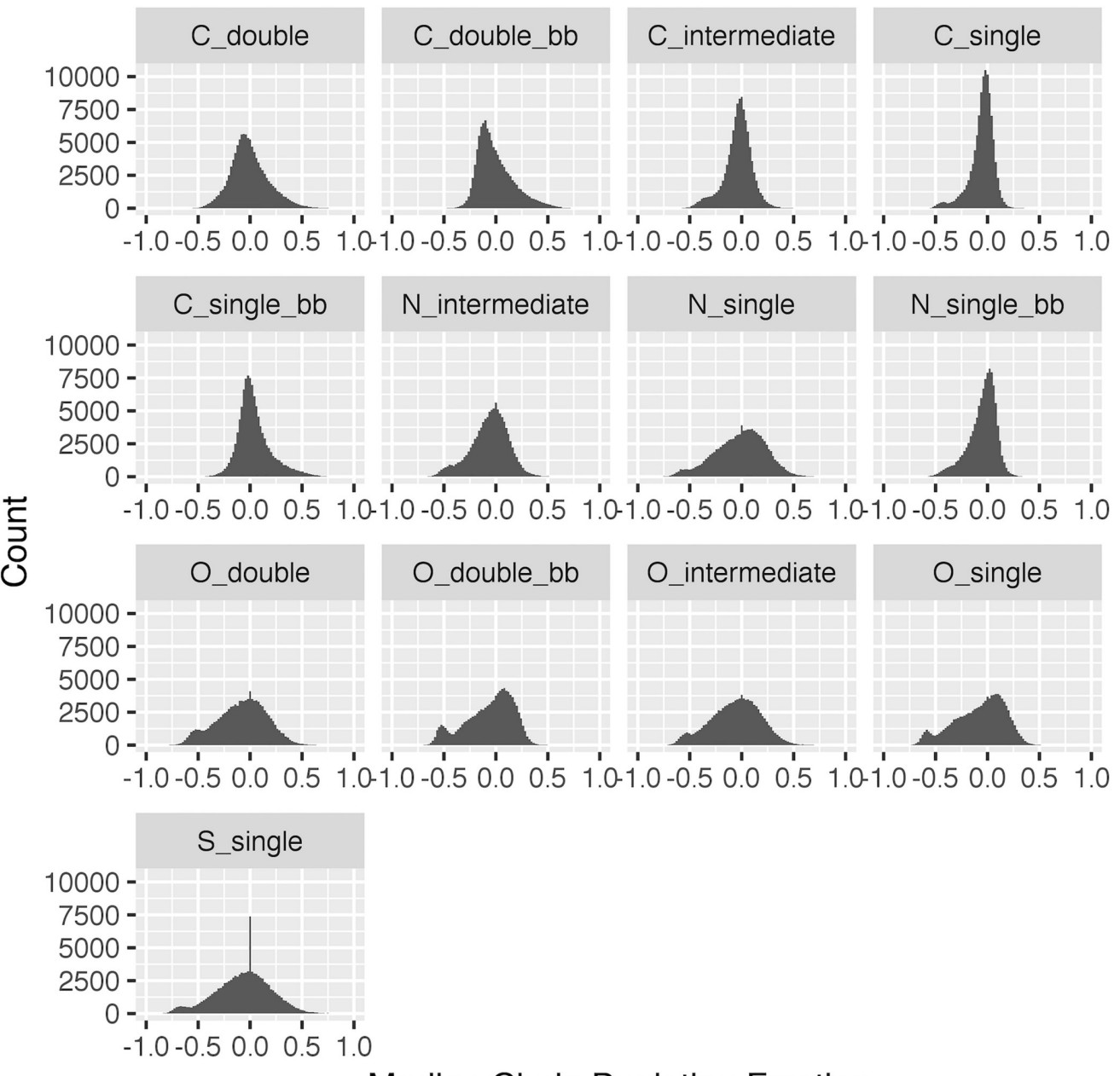

**Fig 7. Histogram of the median chain deviation fraction for all structures in the PDB.**

as random background or water noise. Whereas on panel B, there are some difference density blobs worth about 16 and 29 electrons, which could imply actual missing atoms from the model. Moreover, the red missing electron density and the green extra density suggest that the side chain of A389 glutamine should be modeled at the green mesh position rather than the current position. In a similar manner, regions of interest that are common to many PDB entries can be automatically filtered based on electron deviation quality before systematic analysis.

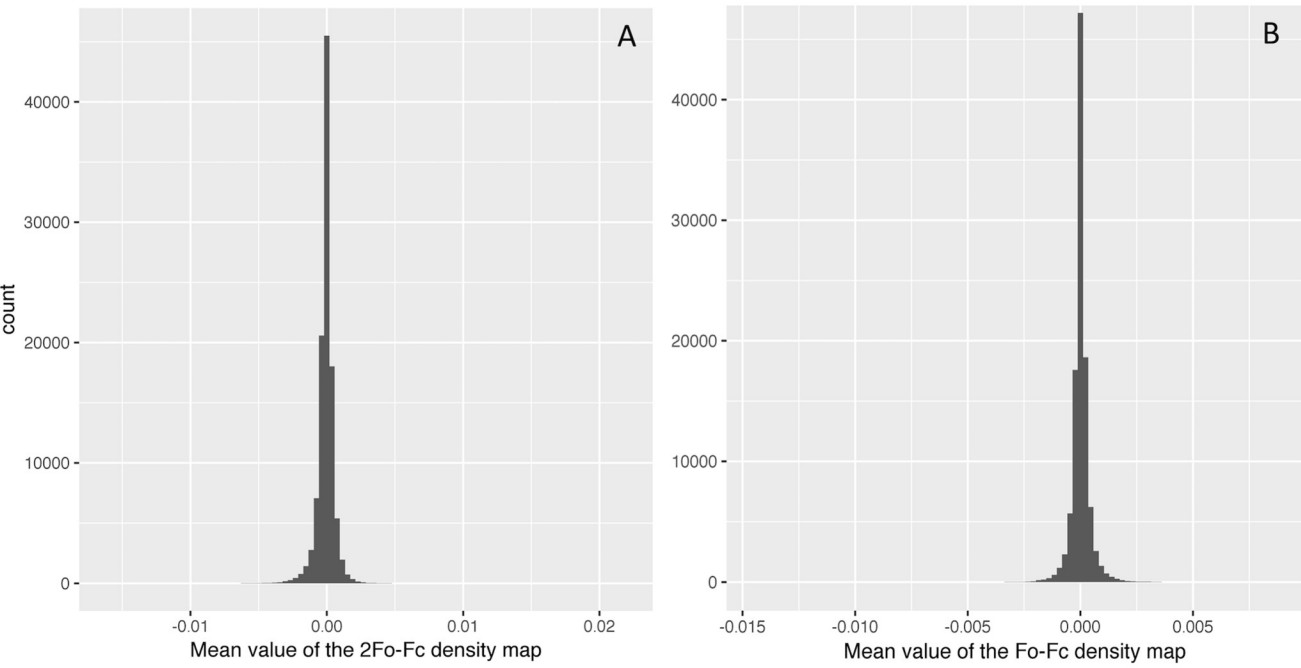

**Fig 8. Histogram of the mean value of electron density maps provided by PDBe.** The histogram illustrates that most of the electron density maps in the PDB are effectively zero-meaned. A) 2Fo-Fc density map, B) Fo-Fc density map.

## Discussion

One of the biggest challenges in using electron density maps is that they are in arbitrary scale with no direct physicochemical meaning. This is partly because of missing the magnitude of the structure factor $F_{000}$, which is generally not needed or reported in structure factor files or electron density maps, as it does not affect standard modeling and visualization procedures. Therefore, to put everything back onto an absolute scale so that it is more meaningful for general scientists, this issue needs to be addressed. An approximation of the $F_{000}$ term can be derived from the structure model; however, it is still incomplete because of unknown solvent

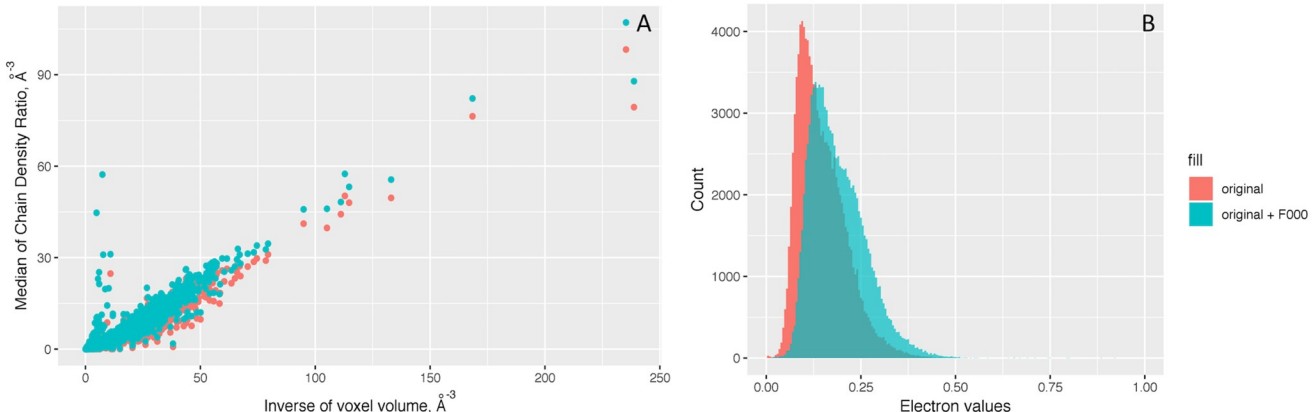

**Fig 9. The absolute scale of the density ratio for all structures in the PDB.** Panel A shows the density ratios vs. inverse of voxel volume plot indicates that there is a consistent 1:3 ratio. Panel B is the histogram of the multiplication of the x and y axes values from Panel A. They both show that the density ratio is not affected much by adding an estimated $F_{000}$ term.

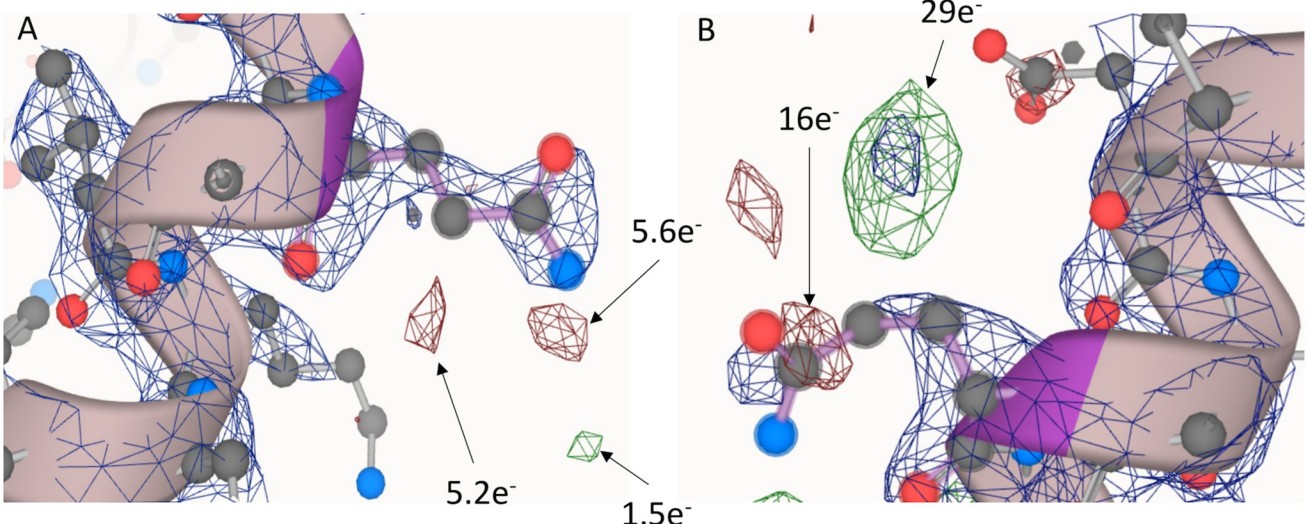

**Fig 10. Evaluative use-case for the electron density conversion factor.** PDB id: 2P7Z, panel A highlighted residue: A.351, panel B highlighted residue: A.389. The blue lattice represents the significant density regions in the 2Fo-Fc map, while the green lattice represents the significant positive discrepant density blobs and the red lattice represents the significant negative discrepant density blobs, both from the Fo-Fc map.

region compositions and potentially other factors, which are calculated in different ways depending on resolution. Also, the zero-mean conversion methods for creation of electron density maps appear to complicate a simple $F_{000}$ correction. This study thus derived new methods that use the median of chain density ratio as a conversion factor to allow the calculation of the missing or excessing electron densities in terms of absolute unit of electrons. These methods implemented in the pdb-eda package provide consistent measures across structures in the PDB with publicly available electron density maps. The pdb-eda methods for deriving a conversion of electron density into a quantity of electrons appear robust with respect to resolution and other PDB entry-specific issues like B-factors, but these methods are currently limited to PDB entries containing a significant peptide/protein component. However, the pdb-eda package contains the basic facilities necessary for deriving atomic radii for other polymeric and repetitive supermacromolecular structures. Also, S1 Fig illustrates a correlation between chain density ratios and resolution, which illuminates a clear path for improvement of atom radii based on resolution. These relationships between atomic radii, B-factors, and resolution with respect to observed electron density have been described previously [35]; however, Fig 5 and S1 Fig provide a useful visualization of these relationships. Also, S1 Fig implies that the current radii are likely optimized for the median resolution of x-ray crystallographic structures present in the wwPDB.

For the purpose of region-specific model evaluation, our pdb-eda package derives a conversion factor from arbitrary electron density per $Å^3$ to electrons per $Å^3$ based only on the provided 2Fo-Fc map and uses this conversion factor to convert electron density discrepancy from the provided Fo-Fc map into absolute units of electrons of discrepancy. At a specified number of standard deviations (sigma level), voxels with significant electron density discrepancies can be detected, absolutely summed, and converted into units of electrons of discrepancy to evaluate local regions [36]. A possibly sophisticated approach would be to use a region-specific estimate of noise [25, 37] to detect voxels with significant electron density discrepancies within a local region.

A complementary method called electron density support for individual atoms (EDIA) utilizes the 2Fo-Fc map to evaluate the electron density support for the location of individual atoms as well as groups of atoms within the electron density [38]. However, the values for EDIA do not have a direct physiochemical interpretation like electrons of discrepancy that our pdb-eda package derives from the Fo-Fc map. But between these two methods, both the Fo-Fc and the 2Fo-Fc maps can be directly used to evaluate the structural quality of a region of interest with respect to the experimental data. Moreover, pdb-eda electrons of discrepancy and EDIA metrics should be highly complementary since the first focuses on the evaluation of significant electron density discrepancies in the Fo-Fc map and EDIA focuses on the evaluation of less significant (below 1.2σ) electron density in the 2Fo-Fc map. Therefore, these new measures provide useful region-specific model evaluation and are suitable for systematic quality control analyses across large numbers of PDB structure entries, as demonstrated for bound metal ion regions [36].

Over time, the user-base of the wwPDB has shifted from mainly protein crystallographers to a broader community of biologists, computational biochemists, and bioinformaticians, which poses new challenges for how structural data is effectively utilized. While crystallographers are familiar with the concept that not all regions in a structure are of the same quality, this concept is relatively unfamiliar to the other scientists, who tend to focus on global metrics of structure quality like resolution, R-factor, and R-free. Moreover, the experimental details are rather overwhelming for non-crystallographers without extensive training. Thus, this study takes advantages of the recent addition of electron density maps to the PDBe, enabling general scientists to better utilize electron density information now available from the public repository. Our Python pdb-eda package provides easy-to-use methods for interpreting and evaluating structural data with a better physiochemical context. The primary goal of this package is to facilitate a shift in x-ray crystallographic structure evaluation from an entry-specific perspective to a region-specific perspective for the broader scientific community that utilizes the PDB.

## Supporting information

**S1 Table. Atom type mapping and the electron counts for the 20 common residues.**
(DOCX)

**S2 Table. Atom-specific electron counts for the 20 common residues.**
(DOCX)

**S1 Fig. Density plot of the median chain deviation fraction for all structures in the PDB of different resolutions.**
(TIFF)

## Acknowledgments

We would like to acknowledge helpful conversations with Dr. David Rodgers.

## Author Contributions

**Conceptualization:** Hunter N. B. Moseley.

**Funding acquisition:** Hunter N. B. Moseley.

**Investigation:** Sen Yao.

**Methodology:** Sen Yao, Hunter N. B. Moseley.

**Software:** Sen Yao.

**Supervision:** Hunter N. B. Moseley.

**Validation:** Hunter N. B. Moseley.

**Visualization:** Sen Yao.

**Writing – original draft:** Sen Yao.

**Writing – review & editing:** Sen Yao, Hunter N. B. Moseley.

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
