## [Decision Letter · Decision Letter 0]

14 May 2020

PONE-D-20-10145

A chemical interpretation of protein electron density maps in the worldwide protein data bank

PLOS ONE

Dear Dr. Moseley,

Thank you for submitting your manuscript to PLOS ONE. After careful consideration, we feel that it has merit but does not fully meet PLOS ONE’s publication criteria as it currently stands. Therefore, we invite you to submit a revised version of the manuscript that addresses the points raised during the review process.

There are a significant numbers of concerns that have to be addressed in the manuscript before it can be accepted. In light of the number and magnitude of the issues raised by the reviewers, the paper will likely undergo a second round of review.

We would appreciate receiving your revised manuscript by Jun 28 2020 11:59PM. To enhance the reproducibility of your results, we recommend that if applicable you deposit your laboratory protocols in protocols.io, where a protocol can be assigned its own identifier (DOI) such that it can be cited independently in the future. For instructions see: http://journals.plos.org/plosone/s/submission-guidelines#loc-laboratory-protocols

We look forward to receiving your revised manuscript.

Kind regards,

Oscar Millet

Academic Editor

PLOS ONE

Journal Requirements:

Reviewers' comments:

Reviewer's Responses to Questions

**Comments to the Author**

1. Is the manuscript technically sound, and do the data support the conclusions?

Reviewer #1: Yes

Reviewer #2: Partly

Reviewer #3: Yes

2. Has the statistical analysis been performed appropriately and rigorously? 

Reviewer #1: Yes

Reviewer #2: No

Reviewer #3: Yes

3. Have the authors made all data underlying the findings in their manuscript fully available?

Reviewer #1: Yes

Reviewer #2: Yes

Reviewer #3: Yes

4. Is the manuscript presented in an intelligible fashion and written in standard English?

Reviewer #1: Yes

Reviewer #2: Yes

Reviewer #3: Yes

5. Review Comments to the Author

Reviewer #1: This is a clear manuscript that describes a process by which electron density maps from protein structures are renormalized to units of number of electrons - which has a direct physical interpretation. It is a valuable contribution to the field of protein science that is suitable for publication in its current format. This reviewer is someone who routinely uses the PDB to obtain region-specific information about proteins and we anticipate exploring the tools described in this article and assessing their suitability for determining the quality of local protein structure regions.

One page six at the bottom - the rho character is misprinted as a >.

I was a little concerned about subsuming the hydrogen electrons into the heavy atom as this would explain variations in the radii of atoms within a structure. It is unclear what would be lost by ignoring these electrons completely.

At the end, some use cases are presented which is helpful and the comment is made that a deviation of 6 electrons is negligible but a dozen or more could represent a problematic region of the structure. At what point is the deviation in number of electrons significant from a physical chemical perspective?

Reviewer #2: General comments

The authors state, very correctly, that there exists a need to be able to study electron density in an independent manner as the election density calculated from structure factors exists on a relative scale. The problem of relative density scales means that it is difficult for large numbers of PDBs to be evaluated collectively. It is also perhaps confusing for scientists, who are not particularly familiar with MX, biologists for example, and they might be drawn into making incorrect conclusions based on a spurious understanding of relative electron density maps. The authors propose, as a solution to these problems, a novel method that not only translates the arbitrary units of electron density into electrons per angstroms cubed, but also, normalises the density. Unlike previous methods, which require structure factors for the F000 calculation, this technique instead is based upon optimised atomic radii which have, in turn, been calculate from observations of the entire PDB.

As a much as I appreciate the goals this work I think this work would be made more powerful it if the authors could more properly highlight the benefits of this method in comparison to an F000 calculation. The authors highlight that there are two groups of people who would benefit from this work: less informed scientists who are viewing single structures, and bioinformaticians, viewing 1000s. I feel that this article could be improved by treating these two goals separately e.g. at a very specific level for a biologist interested in a single enzyme and a bioinformatician looking at 1000s of enzymes. The results could then properly demonstrate how this new method of analysing structures could be implemented and how it can improve the analysis of structures.

Another aspect which I think is slightly unclear/could be improved upon, is a direct comparison with the current method of putting electron density maps on the same scale i.e. from the F000 [Lang et al., (2014)] and by utilizing deposited structure factor data. The authors have not really made clear why this method cannot be automated and done on a larger scale. The authors have attempted to do an F000 calculation but then negated to include a bulk solvent estimation. Lang et al., (2014) does describe how to do this and it, therefore, seems strange that the authors have attempted to compare their method with the F000 calculation without attempting to take into account bulk solvent. This comparison should be made at both the specific (single structure electron density) and the 1000s structure level to really show that the authors’ method is an improvement. The authors’ method is independent of structure factor data and can, therefore, be used on the whole PDB, not just those PDBs which have been deposited since 2008. By my estimation, there should be some 47,569 structures - not inconsiderable - deposited before 2008 where an F000 calculation cannot be performed - the authors could highlight this advantage more fully.

There also seems to be a false premise in the way that the authors have calculated their normalised electron density radii. Part of the calculation appears to be based upon data derived from the PDB as a whole. However, as the authors say in the introduction, the radii deposited in the PDB are all on a relative scale and, therefore, difficult to compare even in the aggregate. Could the authors either confirm that my reading of the article is incorrect and that this has not been done, or if it has, explain that the results they derive are still accurate.

Finally, the referencing in the text is appears to be a bit poor in places. The authors have referenced papers which do not appear to support their conclusions and this distracts from the paper (specific comments are outlined under point 4 below). Please check that the references you have used to cite a particular point are correct.

Is the manuscript technically sound, and do the data support the conclusions? - Partly

As outlined in the general comments, in my opinion the paper could be greatly improved by a clear demonstration of the benefits of using this method in the results section. This could be done by first giving a real comparison with an F000 calculation at both the specific and general level, and then an example of how this method could be used in practice for a particular scientific question.

2. Has the statistical analysis been performed appropriately and rigorously? - No

No statistical analyses appear to have been performed. The authors offer very non-specific comments when comparing how sparse their data are (for example bottom of page 11, top of page 12) and these analyses would be greatly improved by calculating if changes in relative variability are statistically significant.

3. Have the authors made all data underlying the findings in their manuscript fully available? - Yes

I commend the authors for the clarity and ease of accessing their software.

4. Is the manuscript presented in an intelligible fashion and written in standard English? - Yes

Specific comments in text:

Suggest that x-ray should be written as X-ray.

Introduction

Page 3

Bottom-page - Suggestion to correct, “These low-quality regions arise from structural model and electron density mismatches can be due to a variety of reasons including problems with regional protein mobility (8), data processing (9), or model fitting (10).”

To

“These low-quality regions arise from structural model and electron density mismatches that can be due to a variety of reasons including problems with regional protein mobility (8), data processing (9), or model fitting (10).”

Same sentence - check references.

8 and 9 are very specific references and do not obviously support the authors’ statements.

Page 4

Top-page - Suggestion to correct, “Therefore, evaluation…”

to

“Therefore, the evaluation”

Figure 1. The image quality of panel is poor, suggestion to improve resolution of image used for publication. Furthermore, a suggestion to use ‘standard’ structure visualisation programs used by the structural biology community for example PyMOL or Chimera to parallel how structural biologists view and present structures.

Page 5

Mid-page - Many electron density map viewers (15, 16) - 16 does not reference an electron density map viewer. Perhaps add common visualisation tools and their references such at Coot, Pymol and Chimera.

Mid-page - Not clear what is meant by “electrons of density” - do the authors mean “electrons represented by the density”?

Mid-page - Suggestion to correct ”these methods require a reanalysis of the underlying structure factors with software packages that are not easily automated (19, 20).” Suggestion to remove, “that are not easily automated”

Methods

Page 6

Mid-page - Should this be 1.5 and 3 σ (sigma)?

Bottom-page - (>m) should this be ⍴m (rho m)?

Page 7

Top-page - "Since hydrogen is normally not resolvable within electron density maps, their electrons were added to their bonded atom." The authors show later that for high resolution their electron density model does not work as well as for low resolution structures. Can the authors comment on whether high resolution structures need their hydrogen electrons specifically modeled? Instead of being included with the electrons of their bonded atom?

Top-page-second paragraph - Suggestion to correct. “After all atom electron densities are calculated, they are aggregated into residue and chain densities where the residue cloud contains at least 4 atoms and the chain cloud contains at least 50 atoms. The overlapping density voxels between two or more atoms are only counted once through the aggregation. The total number of electrons are calculated by adding contributing atom’s electron numbers together. Residue (rr) and chain (rc) density ratios are then calculated accordingly.”

To

“After all the atom electron densities were calculated, they were aggregated into residue and chain densities where the residue cloud contains at least 4 atoms and the chain cloud contains at least 50 atoms. The overlapping density voxels between two or more atoms were only counted once through the aggregation. The total number of electrons were calculated by adding the contributing atom’s electron numbers together. Residue (rr) and chain (rc) density ratios are then calculated accordingly.

Bottom-page - Suggestion to correct - “…we then define a more universal measure…” to “…we then defined a more universal measure…

Page 8

Optimisation of radii - Suggestion to correct - “After the initial calculation, the median density ratios of different atom types were still quite different from each other. Thus, to achieve a more uniformly interpretable density ratio within a structure as well as across structures, an optimization of radii is performed. First, we tested the radius for each atom type on 100 random structures and obtained an initial estimation of the radii. The metric we used to optimize is the median of corrected chain deviation fraction (fi-corrected) for a given atom type. Based on the results from the initial step, we then optimize one atom type at a time on 1000 randomly selected structures. For every iteration, the atom type that has the largest deviation from last round is optimized. Different radii are tested for the given atom type and the radius that has a median corrected chain deviation fraction closest to zero is picked out. At the end of each optimization, the set of bfactor slopes are updated as well. This process goes on until the median chain deviation fraction for all atom types are smaller than 0.05. The final set of radii are then tested on another 1000 random structures and the whole PDB database for validation.”

to

“After the initial calculation, the median density ratios of different atom types were still quite different from each other. Thus, to achieve a more uniformly interpretable density ratio within a structure as well as across structures, an optimization of radii was performed. First, we tested the radius for each atom type on 100 random structures and obtained an initial estimation of the radii. The metric we used to optimize was the median of corrected chain deviation fraction (fi-corrected) for a given atom type. Based on the results from the initial step, we then optimize one atom type at a time on 1000 randomly selected structures. For every iteration, the atom type that had the largest deviation from last round, was optimized. Different radii were tested for the given atom type and the radius that had a median corrected chain deviation fraction closest to zero was picked out. At the end of each optimization, the set of bfactor slopes was updated as well. This process went on until the median chain deviation fraction for all atom types were smaller than 0.05. The final set of radii were then tested on another 1000 random structures and the whole PDB database for validation.”

Page 9

Can the authors comment on why their F000 calculation does not include a bulk solvent estimation?

Results

Fig. 3 - panels D-F have not been referenced in the text, suggestion to remove or to reference in the text.

Page 11

Fig. 5 - Suggestion to separate graphs further. At present the labels of the X axes overlap and make them challenging to read.

Page 11

Top-page - Please comment on whether the observed improvement is statistically significant.

Table. 1 - For clarity, suggestion to indicate which atom types are side and which are main chain using an additional column in the table.

Page 13

Mid-page - Suggestion to correct “The final set of radii was first tested on another 1000 random structures and all atom types hold true to have no more than a 5% chain deviation fraction. It was then applied to the all PDB structures that has usable electron density data…”

To

“The final set of radii was first tested on another 1000 random structures and all atom types held true to have no more than a 5% chain deviation fraction. It was then applied to the all PDB structures that had usable electron density data…”

Mid-page - Can the authors add a comment here about how they have identified that this model is better or worse for high-resolution data (perhaps with reference to the methods section).

Page 14

Mid-page - Suggestion to correct - “Thus, conversion is not as simple as adding an F000 term as often theoretically represented in textbooks (25, 26)” - Please note that ref 25 is not a textbook and ref 26 does not imply that the bulk solvent calculation is easy. Consider revising this sentence.

Page 15

Fig. 10 - From this figure it is not clear what benefit this annotation would give a non-crystallographer. I agree there is now an estimation of the electron density in terms of the number of electrons. But what does this mean for a biologist? Is it really a great improvement on sigma? For a non-crystallographer, and even many crystallographers, what difference is a number of electrons or a sigma contour level estimation of the density? A cut off in one or the other still feels like an arbitrary rule to live by. To rectify this, suggestion that this figure should demonstrate the value of method. It should perhaps also show the output from the software and indicate how a scientist would find the number of electrons associated with each blob and then why this image/representation would ease interpretation. Could the authors please include such a Figure as a demonstration of the value of this method.

Reviewer #3: Review of PLOS ONE ms PONE-D-20-10145 “A chemical interpretation of protein electron density maps in the worldwide protein data bank” by S.Yao & H.N.B.Moseley

The goal of this work is very commendable (to generate electron density on absolute level) and the solution is elaborated through a convoluted algorithm, which - although basically, I think, correct and well documented - is… unnecessary. In principle, the Authors arrive at the “arbitrary” => “absolute” scale factor by a tedious comparison of the experimental electron density map with its atomic model. Or in other words, they want to scale apples with oranges. But a much simpler solution exists: compare oranges with oranges. In the simpler approach one would first convert the atomic model to its Fourier Transform Fc(hkl), and then use a subset of those calculated structure factors corresponding exactly to the set of experimental Fo(hkl) data, to generate ρc(xyz), i.e. the calculated (Fc) electron density map. This map would be the target for scaling the experimental (observed) ρo(xyz) electron density map (Fo). The scaling would be of course linear: ρc(xyz) = a·ρo(xyz) + b, and would have to be fulfilled at all grid points on which the two maps have been calculated. Since this strictly linear problem is hugely overdetermined, only the most reliable grid points could be included, e.g. within 1 A of the atomic centers of the model, or indeed within the atomic radii used in the ms. The set of linear equations could be solved by the method of least squares, with the addition of a robust method to filter out outliers, e.g. if some fragments of the model are of poor quality. This way the best values of the a and b parameters are obtained. I have not tried this algorithm myself - it’s not my paper. But I am pretty sure it should work quite well. At least the Authors should try a simple method first, before proposing an algorithm that may be unduly complicated.

In conclusion, I cannot recommend acceptance of this paper in its present form. In addition to the doubts outlined above, there is one more misgiving. Assuming that we have electron density re-scaled to absolute units by one method or the other, the real question is : “So what?” The Authors should provide examples to clearly demonstrate what is possible with their maps that would not be possible with sigma-scaled 2Fo-Fc and Fo-Fc maps. Right now there is a lot of verbal promise but very little of concrete proof. BTW, the caption of Fig. 10 is amazingly unhelpful.

It may be irrelevant in view of my final recommendation (reject), but since I’ve read the paper very carefully, I’m also including my more specific comments and critical remarks, divided into “Substantive” and “Technical”.

Substantive problems

The convoluted algorithm includes a series of corrections, one after another. At some point the reader is lost as to the purpose of all those corrections. Perhaps it would be helpful to add a graph showing the distribution of the sample 1000 structures?

The Authors should calibrate their method with ultrahigh-resolution PDB crystal structures which provide accurate estimate of electron density levels in e/A3 because they have a large Ewald sphere of very accurate F(hkl) data and in addition allow reliable estimation of F(000) since the (nearly) complete atomic content of the unit cell is practically known. The recommended examples would be the PDB entries 3NIR (0.48 A, crambin, highest resolution but unsatisfactory refinement), 1EJG (0.54 A, crambin) (comparison of the two crambin structures would provide an interesting “internal standard”) and 2VB1 (0.65 A, lysozyme) for proteins, and 3P4J (0.55 A, Z-DNA) for nucleic acids.

I also note that, infrequent as they are, occasionally ultrahigh-resolution macromolecular structures are presented with the electron density maps expressed on the absolute scale (e.g. Addlagatta et al. (2001) Acta Cryst. D57: 649-663). Such maps could also be used to validate the scaling procedures proposed in this work.

p3, “These low-quality regions arise from structural model and electron density mismatches…”, actually, most often the low-quality regions arise because of absence of electron density.

p6, I am surprised that the Authors have completely overlooked the paper by Tickle (2012) Acta Cryst. D68: 454-467, which is the standard classical reference for sigma-contoured electron density maps and more. Also, there is a good discussion there about the radii of atoms that cover 95% of electron density, and about the influence of B-factors and resolution (see 5.6. The limiting radius of the atomic density; as well as Table 3 and Fig. 11 therein).

Fig. 2 caption; it is correct to include H atoms in the electronic inventory of their “carriers” if H atoms were not included in Fc calculations. However, as is very often the case, if H atoms were included in Fc, then this strategy is incorrect. Moreover, if the electron density of H atoms is added to the bound atom, then interpretation by a simple spherical volume around that atom will lead to systematic errors. In addition, the resolution of the electron density map is also important.

I don’t understand the idea of dividing the formula for F(000) in eq. (6) by V. Structure factors F (including F(000), of course), are expressed directly in electrons (e). Also, counting the electron contribution of the solvent molecules is necessary, but in most cases we miss a lot of water molecules (not included in the model). At low resolution, we cannot count the water molecules at all. What could be done, however, would be to estimate the number of water molecules from Matthews volume and specific density of water (1 g/cm3) and add their electrons to F(000).

I wonder, how much the re-scaled difference maps show-cased on p15 and Fig. 10 would differ from normal mFo-DFc and 2mFo-DFc maps. I think such a comparison should be illustrated.

The method should be extended (in the future?) to explicitly apply to nucleic acids as well. Nucleic acids constitute an important segment of PDB structures.

To be of general applicability, a method like this should produce electron density maps in mtz or other ccp4/coot/pymol-readable format, so that they could be easily loaded and displayed for visualization.

Technical remarks

p3, “activities happens”, singl./pl. problem.

p6 and elsewhere, the Greek letter sigma (and other symbols) is misprinted as a funny character.

I couldn’t find anywhere in the ms the key information about the density of the grid over which the electron density is calculated and analyzed.

Fig. 2 caption, “20 common amino acid”, singl./pl. problem.

Correct the grammar in Fig. 2 (“by atom the b-factor”).

p7, “total number of electrons are”, singl./pl. problem.

On p8, the term “chain deviation fraction” is suddenly changed to “chain fraction” (without definition).

p8 and elsewhere, “on 100 random structures”, change to “of…”.

p10, “Fig. 3. Sina plots of density ratio for atoms, residues, and chains”, perhaps it would be helpful to add a short description a sina plots, for example as in https://ggforce.data-imaginist.com/reference/geom_sina.html: “Sina plot is an enhanced jitter strip chart, where the width of the jitter is controlled by the density distribution of the data within each class.”.

Fig. 3, frankly, I don’t see much difference between the top and bottom panels… Moreover, it seems to me that the x axis of panels C and F shows residues, not chain IDs; I am confused…

The “optimized” atomic radii are usually >= the “original” radii. An outstanding exception in Table 1 is S. Any reason why?

p13, last sentence before new section, don’t begin sentence with “And…”.

p13, explain “distribution modes”.

p13, “It was then applied to the all PDB structures that has usable electron density data”, several grammatical errors in this sentence.

Fig. 8 caption talks about 2mFo-DFc maps, but panels A/B are supposed to correspond to 2Fo Fc/Fo Fc maps.

Something wrong with the first sentence of Fig. 9 caption.

p14, “with a local region.”, perhaps “within a local region.”.

p15, difference electron density of “16 and 29 e” sounds like serious overshooting. Should be metal or halide ions, or at least S.

Fig. S1, the axes need proper labels and units. The values of the ticks on the x axes are squeezed too much and unreadable. The plots, with the current caption, are not very useful.

Tables S1 and S2 with the inventory of bonds and atoms in standard amino acid residues are banal and could be omitted.

Please note that the proper symbol to be used for Atomic Displacement Parameter (temperature factor) is B-factor.

I am glad to note that the GITHUB documentation and examples seem to work well. I note that GITHUB reports version 1.0.1. and PythonPackageIndex version 1.1.0. There are also some shortcomings and bugs. For example, the command:

python3 -m pdb_eda single 3han 3han.all.csv --all --out-format=csv

instead of human-readable printouts, returns a printout of rather useless python objects.

Execution of:

python3 -m pdb_eda single 3han 3han.all.csv --all --out-format=csv

generates an error message:

AttributeError: 'int' object has no attribute 'aggregateCloud'

which is not very helpful.

6. PLOS authors have the option to publish the peer review history of their article (what does this mean?). If published, this will include your full peer review and any attached files.

Reviewer #1: No

Reviewer #2: Yes: John H. Beale

Reviewer #3: No

---

## [Author Response · Author response to Decision Letter 0]

27 Jun 2020

Reviewer #1: 

This is a clear manuscript that describes a process by which electron density maps from protein structures are renormalized to units of number of electrons - which has a direct physical interpretation. It is a valuable contribution to the field of protein science that is suitable for publication in its current format. This reviewer is someone who routinely uses the PDB to obtain region-specific information about proteins and we anticipate exploring the tools described in this article and assessing their suitability for determining the quality of local protein structure regions.

Response:

We thank the reviewer for recognizing the value of our research presented here. We are trying to provide tools that make it easier to evaluate structural regions without needing to work with structure factor files and x-ray crystallography software that requires a lot of expertise to use properly. 

Issue 1:

One page six at the bottom - the rho character is misprinted as a >.

Response:

Fixed.

Issue 2:

I was a little concerned about subsuming the hydrogen electrons into the heavy atom as this would explain variations in the radii of atoms within a structure. It is unclear what would be lost by ignoring these electrons completely.

Response:

The hydrogen electron is part of the electron density and cannot be ignored. Exactly how to represent the electron cloud around the heavy atom and the hydrogen could be an issue if the resolution of a given structure was really high (probably below 0.5 angstroms). But pragmatically, macromolecular x-ray crystallographic structures are not high enough resolution to make a difference.

Issue 3:

At the end, some use cases are presented which is helpful and the comment is made that a deviation of 6 electrons is negligible but a dozen or more could represent a problematic region of the structure. At what point is the deviation in number of electrons significant from a physical chemical perspective?

Response:

We do not know how much of a deviation is problematic. It will depend on how the region is being used and to what level of interpretation. We are actively exploring how quantified regional deviations affect molecular docking of ligands, but have just started this collaborative work.

Reviewer #2: 

The authors state, very correctly, that there exists a need to be able to study electron density in an independent manner as the election density calculated from structure factors exists on a relative scale. The problem of relative density scales means that it is difficult for large numbers of PDBs to be evaluated collectively. It is also perhaps confusing for scientists, who are not particularly familiar with MX, biologists for example, and they might be drawn into making incorrect conclusions based on a spurious understanding of relative electron density maps. The authors propose, as a solution to these problems, a novel method that not only translates the arbitrary units of electron density into electrons per angstroms cubed, but also, normalises the density. Unlike previous methods, which require structure factors for the F000 calculation, this technique instead is based upon optimised atomic radii which have, in turn, been calculate from observations of the entire PDB.

Response:

We thank the reviewer for recognizing the importance of the problem we are trying to address in this research.

Issue 1:

As a much as I appreciate the goals this work I think this work would be made more powerful it if the authors could more properly highlight the benefits of this method in comparison to an F000 calculation. The authors highlight that there are two groups of people who would benefit from this work: less informed scientists who are viewing single structures, and bioinformaticians, viewing 1000s. I feel that this article could be improved by treating these two goals separately e.g. at a very specific level for a biologist interested in a single enzyme and a bioinformatician looking at 1000s of enzymes. The results could then properly demonstrate how this new method of analysing structures could be implemented and how it can improve the analysis of structures.

Response:

If it was practical to calculate an F000 for thousands of PDB entries, we would have done this and more importantly, there would not be a need for the methods and software that we have developed. Therefore, we presented alternative ways to evaluate and validate the method. While these approaches do not represent a classical analytical cross-validation with another method, they do represent a statistical validation that demonstrates mostly unimodal distributions that highlight the level of uncertainty in the data. Please understand that there are different approaches to validation. The one we have presented is statistically rigorous and compelling. The corresponding author, Hunter Moseley, has degrees in Chemistry, Biochemistry, Mathematics, and Computer Science. He currently teaches statistics to biomedical graduate students and authored the statistics for biologists introduction in Bioinformatics: A Practical Guide to the Analysis of Genes and Proteins 4th Edition, editors: Andreas D. Baxevanis, Gary D. Bader, and David S. Wishart, John Wiley & Sons: New Jersey (2020). We understand that there is often confusion between analytical and statistical approaches, but rigorous evaluation is possible with both types of approaches. We often use both types of approaches to validate our methods and research, when both can be utilized. 

Issue 2:

Another aspect which I think is slightly unclear/could be improved upon, is a direct comparison with the current method of putting electron density maps on the same scale i.e. from the F000 [Lang et al., (2014)] and by utilizing deposited structure factor data. The authors have not really made clear why this method cannot be automated and done on a larger scale. The authors have attempted to do an F000 calculation but then negated to include a bulk solvent estimation. Lang et al., (2014) does describe how to do this and it, therefore, seems strange that the authors have attempted to compare their method with the F000 calculation without attempting to take into account bulk solvent. This comparison should be made at both the specific (single structure electron density) and the 1000s structure level to really show that the authors’ method is an improvement. The authors’ method is independent of structure factor data and can, therefore, be used on the whole PDB, not just those PDBs which have been deposited since 2008. By my estimation, there should be some 47,569 structures - not inconsiderable - deposited before 2008 where an F000 calculation cannot be performed - the authors could highlight this advantage more fully.

Response:

We have not automated analysis of structure factors, because we focused on the Fo-Fc maps. If you can identify a method that automates use of structure factors to derive F000, we are very willing to do a direct comparison. But ultimately, our goal was to analyze the Fo-Fc and 2Fo-Fc maps for regional analysis. Therefore, our development efforts were focused there. We went through the old CCP4 codebase to determine the ccp4 file formats and implemented libraries to fully parse these files. To produce the quality of the codebase we have implemented takes a significant amount of effort. We are not just developing methodology, but implementing it in high-quality code bases following best software development practices. This is in contrast to the vast majority of scientific software that is developed and published.

Also, our method cannot be used on the whole wwPDB. There are many entries that do not have Fo-Fc electron density maps made available by PDB in Europe. So, no we cannot claim that our methods work on all x-ray crystallographic structures deposited in the wwPDB.

Issue 3:

There also seems to be a false premise in the way that the authors have calculated their normalised electron density radii. Part of the calculation appears to be based upon data derived from the PDB as a whole. However, as the authors say in the introduction, the radii deposited in the PDB are all on a relative scale and, therefore, difficult to compare even in the aggregate. Could the authors either confirm that my reading of the article is incorrect and that this has not been done, or if it has, explain that the results they derive are still accurate.

Response:

Actually, we started with radii published in the literature: 

“To calculate the total electron density around each atom, we initially used the radii from literature (23) and calculated the sum of all densities within the corresponding radius.”

Then we optimized these radii utilizing a set of 1000 randomly selected PDB entries, then tested against another set of 1000 randomly selected PDB entries, and ultimately validated them against all usable wwPDB entries.

Issue 4:

Finally, the referencing in the text is appears to be a bit poor in places. The authors have referenced papers which do not appear to support their conclusions and this distracts from the paper (specific comments are outlined under point 4 below). Please check that the references you have used to cite a particular point are correct.

Response:

Thanks for the important feedback. We have updated and improved the references.

Issue 5:

Is the manuscript technically sound, and do the data support the conclusions? - Partly

As outlined in the general comments, in my opinion the paper could be greatly improved by a clear demonstration of the benefits of using this method in the results section. This could be done by first giving a real comparison with an F000 calculation at both the specific and general level, and then an example of how this method could be used in practice for a particular scientific question.

Response:

We understand the reviewer’s desire for an analytical cross-validation comparison with another way to calculate the results, but that was not practical in this instance and is actually a major reason for the development of this approach.

Our other publication where we apply our methods to a systematic analysis of all metal-bound ions across the wwPDB should demonstrate a practical application of our methods: Sen Yao and Hunter N.B. Moseley. "Finding high-quality metal ion-centric regions across the worldwide Protein Data Bank" Molecules 24, 3179 (2019).

Issue 6:

2. Has the statistical analysis been performed appropriately and rigorously? - No

No statistical analyses appear to have been performed. The authors offer very non-specific comments when comparing how sparse their data are (for example bottom of page 11, top of page 12) and these analyses would be greatly improved by calculating if changes in relative variability are statistically significant.

Response:

We used a visualization approach to demonstrate statistical rigor. Figure 5 shows a clear correlation between log B-factors and chain deviation fraction. More importantly, Figure 6 shows the improvement in the density ratio distributions derived for each atom type. This includes a reduction in variation, but more importantly, the improvement in the consistency of these unimodal distributions across atom types. We could have quantified this improvement by calculating the overlap across the distributions with each refinement, but honestly the visualization is just so compelling. In our opinion, a well-done visualization of the distributions so that they can be easily compared (i.e. the reason for the Sina plots in Figure 6), allows for much better interpretation. However, we recognize the utility in providing a quantitation on the improvement in the density ratio distributions. Therefore, we have added this in terms of a standard deviation for each refinement illustrated in Figure 6. In the main text, we highlight this improvement in standard deviation as follows:

“Moreover, the standard deviation of the overall distribution drops to 0.09, representing a 40% decrease in the calculated density ratio variability.”

Issue 7:

3. Have the authors made all data underlying the findings in their manuscript fully available? - Yes

I commend the authors for the clarity and ease of accessing their software.

Response:

Thank you. We put a lot of effort into making our software accessible and usable.

Issue 8:

4. Is the manuscript presented in an intelligible fashion and written in standard English? - Yes

Response:

Thank you. This manuscript has gone through many rounds of revision.

Issue 9:

Specific comments in text:

Suggest that x-ray should be written as X-ray.

Response:

We do not think this is appropriate. The word “x-ray” is not a proper noun.

Issue 10:

Introduction

Page 3

Bottom-page - Suggestion to correct, “These low-quality regions arise from structural model and electron density mismatches can be due to a variety of reasons including problems with regional protein mobility (8), data processing (9), or model fitting (10).”

To

“These low-quality regions arise from structural model and electron density mismatches that can be due to a variety of reasons including problems with regional protein mobility (8), data processing (9), or model fitting (10).”

Response:

Thanks! Fixed:

“These low-quality regions arise from structural model and electron density mismatches that can be due to a variety of reasons including problems with regional protein mobility that can often lead to an apparent lack of electron density [8-10], data processing [11, 12], or model fitting [10, 13, 14].”

Issue 11:

Same sentence - check references.

8 and 9 are very specific references and do not obviously support the authors’ statements.

Response:

These references are specific, but are meant as examples of each issue. We have added additional references that include reviews that highlight these issues.

Issue 12:

Page 4

Top-page - Suggestion to correct, “Therefore, evaluation…”

to

“Therefore, the evaluation”

Response:

Fixed.

Issue 13:

Figure 1. The image quality of panel is poor, suggestion to improve resolution of image used for publication. Furthermore, a suggestion to use ‘standard’ structure visualisation programs used by the structural biology community for example PyMOL or Chimera to parallel how structural biologists view and present structures.

Response:

We have improved the resolution of the image.

Issue 13:

Page 5

Mid-page - Many electron density map viewers (15, 16) - 16 does not reference an electron density map viewer. Perhaps add common visualisation tools and their references such at Coot, Pymol and Chimera.

Response:

Thank you for catching reference 16. This was accidentally added at some point in the revision process. We have added references for Coot, Pymol, and Chimera:

“Many electron density map viewers [19-22] exist for manually examining the quality of a model versus its electron density; however, this software and evaluation approach is not suitable for batch analysis of hundreds of structures.”

Issue 14:

Mid-page - Not clear what is meant by “electrons of density” - do the authors mean “electrons represented by the density”?

Response:

This is exactly what we meant and have modified the text accordingly:

“But this zero-sum representation can be detrimental for understanding a model, especially a local region of a model, where the number of electrons represented by the density or density discrepancy would be useful for evaluation.”

Issue 15:

Mid-page - Suggestion to correct ”these methods require a reanalysis of the underlying structure factors with software packages that are not easily automated (19, 20).” Suggestion to remove, “that are not easily automated”

Response:

This point emphasizes a major advantage that our software provides. But we understand why the reviewer may take offense to the statement, which can be interpreted to mean something else. Therefore, we have reworded it as follow:

“While methods exist that can derive electron density maps on an absolute scale, these methods require a reanalysis of the underlying structure factors with software packages that are not designed for automated use across large numbers of structural entries (19, 20).” 

Issue 16:

Methods

Page 6

Mid-page - Should this be 1.5 and 3 σ (sigma)?

Response:

Thanks for catching this. Looks like some of our Greek symbols got converted to weird symbols going back and forth between Windows and Mac versions of Word. Fixed.

Issue 17:

Bottom-page - (>m) should this be ⍴m (rho m)?

Response:

Same Mac-Windows conversion issue. Fixed.

Issue 18:

Page 7

Top-page - "Since hydrogen is normally not resolvable within electron density maps, their electrons were added to their bonded atom." The authors show later that for high resolution their electron density model does not work as well as for low resolution structures. Can the authors comment on whether high resolution structures need their hydrogen electrons specifically modeled? Instead of being included with the electrons of their bonded atom?

Response:

Figure S1 shows the opposite of what the reviewer interpreted. Our method works a bit better for high resolution structures than low resolution structures. Simply compare the >3 angstroms distribution to the <=1.5 angstroms distribution across the atom types. So no, we do not need to specifically model hydrogen electrons. We expect the resolution would need to be below 0.5 angstroms before this becomes a factor. Because we see a correlation between resolution and the median chain deviation fraction for some atom types, we see a path to a future refinement of the method. Also, we think the current radii are optimized for the median resolution of x-ray crystallographic structures in the wwPDB. We have added a statement to make this point:

“This figure also implies that the current radii are likely optimized for the median resolution of x-ray crystallographic structures present in the wwPDB.”

Issue 19:

Top-page-second paragraph - Suggestion to correct. “After all atom electron densities are calculated, they are aggregated into residue and chain densities where the residue cloud contains at least 4 atoms and the chain cloud contains at least 50 atoms. The overlapping density voxels between two or more atoms are only counted once through the aggregation. The total number of electrons are calculated by adding contributing atom’s electron numbers together. Residue (rr) and chain (rc) density ratios are then calculated accordingly.”

To

“After all the atom electron densities were calculated, they were aggregated into residue and chain densities where the residue cloud contains at least 4 atoms and the chain cloud contains at least 50 atoms. The overlapping density voxels between two or more atoms were only counted once through the aggregation. The total number of electrons were calculated by adding the contributing atom’s electron numbers together. Residue (rr) and chain (rc) density ratios are then calculated accordingly.

Response:

Respectfully, we disagree. We believe that the present verb tense is the correct verb tense to use here. We had used a past verb tense two sentences previous, because it referred to a decision on how to represent the hydrogen electrons.

Issue 20:

Bottom-page - Suggestion to correct - “…we then define a more universal measure…” to “…we then defined a more universal measure…

Response:

Again, we respectfully disagree. We are using a present verb tense in describing the calculations. If this was a specific calculation performed, we would have used the past tense.

Issue 21:

Page 8

Optimisation of radii - Suggestion to correct - “After the initial calculation, the median density ratios of different atom types were still quite different from each other. Thus, to achieve a more uniformly interpretable density ratio within a structure as well as across structures, an optimization of radii is performed. First, we tested the radius for each atom type on 100 random structures and obtained an initial estimation of the radii. The metric we used to optimize is the median of corrected chain deviation fraction (fi-corrected) for a given atom type. Based on the results from the initial step, we then optimize one atom type at a time on 1000 randomly selected structures. For every iteration, the atom type that has the largest deviation from last round is optimized. Different radii are tested for the given atom type and the radius that has a median corrected chain deviation fraction closest to zero is picked out. At the end of each optimization, the set of bfactor slopes are updated as well. This process goes on until the median chain deviation fraction for all atom types are smaller than 0.05. The final set of radii are then tested on another 1000 random structures and the whole PDB database for validation.”

to

“After the initial calculation, the median density ratios of different atom types were still quite different from each other. Thus, to achieve a more uniformly interpretable density ratio within a structure as well as across structures, an optimization of radii was performed. First, we tested the radius for each atom type on 100 random structures and obtained an initial estimation of the radii. The metric we used to optimize was the median of corrected chain deviation fraction (fi-corrected) for a given atom type. Based on the results from the initial step, we then optimize one atom type at a time on 1000 randomly selected structures. For every iteration, the atom type that had the largest deviation from last round, was optimized. Different radii were tested for the given atom type and the radius that had a median corrected chain deviation fraction closest to zero was picked out. At the end of each optimization, the set of bfactor slopes was updated as well. This process went on until the median chain deviation fraction for all atom types were smaller than 0.05. The final set of radii were then tested on another 1000 random structures and the whole PDB database for validation.”

Response:

We agree completely with the verb tense change. Here is the revised section:

“After the initial calculation, the median density ratios of different atom types were still quite different from each other. Thus, to achieve a more uniformly interpretable density ratio within a structure as well as across structures, an optimization of radii was performed. First, we tested the radius for each atom type on 100 random structures and obtained an initial estimation of the radii. The metric we used to optimize was the median of corrected chain deviation fraction (fi-corrected) for a given atom type. Based on the results from the initial step, we then optimized one atom type at a time on 1000 randomly selected structures. For every iteration, the atom type that has the largest deviation from last round was optimized. Different radii were tested for the given atom type and the radius that has a median corrected chain deviation fraction closest to zero was picked out. At the end of each optimization, the set of B-factor slopes were updated as well. This process continued until the median chain deviation fraction for all atom types were smaller than 0.05. The final set of radii were then tested on another 1000 random structures and the whole PDB database for validation.”

Issue 22:

Page 9

Can the authors comment on why their F000 calculation does not include a bulk solvent estimation?

Response:

We think it is because we tried to derive this from the 2mFo-DFc map instead of the structure factors themselves, but honestly we are not sure. We tried a lot of variations and none of them worked.

Issue 23:

Results

Fig. 3 - panels D-F have not been referenced in the text, suggestion to remove or to reference in the text.

Response:

We added the following sentence:

“This improvement percolates through to the residue and chain level, but is not as obvious (see Fig 3, Panels D-F).”

Issue 24:

Page 11

Fig. 5 - Suggestion to separate graphs further. At present the labels of the X axes overlap and make them challenging to read.

Response:

We have fixed this.

Issue 25:

Page 11

Top-page - Please comment on whether the observed improvement is statistically significant.

Response:

We are assuming that the reviewer is referring to the improvements visualized in Figure 6. But, statistical significance generally refers to null hypothesis testing. There is no null hypothesis being testing here. We can quantify the improvement and have done so in terms of a standard deviation of the density ratio values across all atom types at each stage of refinement. So we can quantify the improvement, but whether this is enough to affect statistical significance is context specific and depends on the specific dataset being analyzed. In the main text, we highlight this improvement as follows:

“Moreover, the standard deviation of the overall distribution drops to 0.09, representing a 40% decrease in the calculated density ratio variability.”

Issue 26:

Table. 1 - For clarity, suggestion to indicate which atom types are side and which are main chain using an additional column in the table.

Response:

Backbone atoms are indicated by “_bb”. We have added a footnote to the table to make this clear.

Issue 27:

Page 13

Mid-page - Suggestion to correct “The final set of radii was first tested on another 1000 random structures and all atom types hold true to have no more than a 5% chain deviation fraction. It was then applied to the all PDB structures that has usable electron density data…”

To

“The final set of radii was first tested on another 1000 random structures and all atom types held true to have no more than a 5% chain deviation fraction. It was then applied to the all PDB structures that had usable electron density data…”

Response:

Thanks. We made this change:

“The final set of radii was first tested on another 1000 random structures and all atom types held true to have no more than a 5% chain deviation fraction. It was then applied to all PDB structures that had usable electron density data, and the results are shown in Fig 7.”

Issue 28:

Mid-page - Can the authors add a comment here about how they have identified that this model is better or worse for high-resolution data (perhaps with reference to the methods section).

Response:

In Figure S1, simply compare the >3 angstroms distribution to the <=1.5 angstroms distribution across the atom types. Our method works a bit better for high resolution structures than low resolution structures. More importantly, we think the current radii are optimized for the median resolution of x-ray crystallographic structures in the wwPDB. We have added a statement to make this point:

“This figure also implies that the current radii are likely optimized for the median resolution of x-ray crystallographic structures present in the wwPDB.”

Issue 29:

Page 14

Mid-page - Suggestion to correct - “Thus, conversion is not as simple as adding an F000 term as often theoretically represented in textbooks (25, 26)” - Please note that ref 25 is not a textbook and ref 26 does not imply that the bulk solvent calculation is easy. Consider revising this sentence.

Response:

The way the reviewer interpreted this sentence was not our intended meaning. We have reworded the sentence as follows:

“Thus, an F000 term as theoretically represented in textbooks and papers (25, 26) is not easy to calculate and likely depends on software parameters used in the creation of the map.“ 

Issue 30:

Page 15

Fig. 10 - From this figure it is not clear what benefit this annotation would give a non-crystallographer. I agree there is now an estimation of the electron density in terms of the number of electrons. But what does this mean for a biologist? Is it really a great improvement on sigma? For a non-crystallographer, and even many crystallographers, what difference is a number of electrons or a sigma contour level estimation of the density? A cut off in one or the other still feels like an arbitrary rule to live by. To rectify this, suggestion that this figure should demonstrate the value of method. It should perhaps also show the output from the software and indicate how a scientist would find the number of electrons associated with each blob and then why this image/representation would ease interpretation. Could the authors please include such a Figure as a demonstration of the value of this method.

Response:

We are trying to improve the chemical interpretation of the deviation. Biological interpretation is another step after the chemical or biochemical interpretation. We demonstrate the utility of our method and software in the following paper: Sen Yao and Hunter N.B. Moseley. "Finding high-quality metal ion-centric regions across the worldwide Protein Data Bank" Molecules 24, 3179 (2019).

Reviewer #3: 

Review of PLOS ONE ms PONE-D-20-10145 “A chemical interpretation of protein electron density maps in the worldwide protein data bank” by S.Yao & H.N.B.Moseley

The goal of this work is very commendable (to generate electron density on absolute level) and the solution is elaborated through a convoluted algorithm, which - although basically, I think, correct and well documented - is… unnecessary. In principle, the Authors arrive at the “arbitrary” => “absolute” scale factor by a tedious comparison of the experimental electron density map with its atomic model. Or in other words, they want to scale apples with oranges. But a much simpler solution exists: compare oranges with oranges. In the simpler approach one would first convert the atomic model to its Fourier Transform Fc(hkl), and then use a subset of those calculated structure factors corresponding exactly to the set of experimental Fo(hkl) data, to generate ρc(xyz), i.e. the calculated (Fc) electron density map. This map would be the target for scaling the experimental (observed) ρo(xyz) electron density map (Fo). The scaling would be of course linear: ρc(xyz) = a·ρo(xyz) + b, and would have to be fulfilled at all grid points on which the two maps have been calculated. Since this strictly linear problem is hugely overdetermined, only the most reliable grid points could be included, e.g. within 1 A of the atomic centers of the model, or indeed within the atomic radii used in the ms. The set of linear equations could be solved by the method of least squares, with the addition of a robust method to filter out outliers, e.g. if some fragments of the model are of poor quality. This way the best values of the a and b parameters are obtained. I have not tried this algorithm myself - it’s not my paper. But I am pretty sure it should work quite well. At least the Authors should try a simple method first, before proposing an algorithm that may be unduly complicated.

Response:

We do not argue that working with structure factors could allow one to derive a conversion factor with a simpler approach. We are demonstrating a method to do this from the Fo-Fc maps that is implemented is a relatively easy to use software package. Complicated or not, we strove to demonstrate that it works.

Honestly, the third reviewer is a crystallographer and not the target audience for our software. Any well-trained crystallographer is going to expertly use their favorite raw data analysis software to derive and evaluate new macromolecular structures. But we want to enable the non-crystallographer users of the PDB to be able to make better choices in the PDB entries that they use and not just base it on the resolution, R-factor, and R-free of the entry. Also, we are trying to provide easy-to-use software that can be applied to large numbers of PDB entries so that systemic analyses can be performed using easy to obtain processed x-ray crystallographic data (Fo-Fc and 2Fo-Fc maps) provided by the PDB in Europe REST interface.

Issue 1:

In conclusion, I cannot recommend acceptance of this paper in its present form. In addition to the doubts outlined above, there is one more misgiving. Assuming that we have electron density re-scaled to absolute units by one method or the other, the real question is : “So what?” The Authors should provide examples to clearly demonstrate what is possible with their maps that would not be possible with sigma-scaled 2Fo-Fc and Fo-Fc maps. Right now there is a lot of verbal promise but very little of concrete proof. BTW, the caption of Fig. 10 is amazingly unhelpful.

Response:

To reject a manuscript presenting a method that the reviewer concludes does work is not following the PLOS One review criteria that says to review on the grounds of if the presented research is sound and done correctly. If we have presented a method that works and is scientifically sound, then rejecting because the reviewer does not see a personal use of the research is not a valid reason to reject for this journal.

Furthermore, we have demonstrated an application of this methodology to the systematic characterization of metal-binding sites across the wwPDB: Sen Yao and Hunter N.B. Moseley. "Finding high-quality metal ion-centric regions across the worldwide Protein Data Bank" Molecules 24, 3179 (2019). We are looking at applying this method as a way to improve molecular docking of ligands, but that work has just started.

We apologize that we may have developed a software package that the reviewer does not personally find useful to their research. But in all fairness, we were not developing our software package for crystallographers.

Issue 1:

It may be irrelevant in view of my final recommendation (reject), but since I’ve read the paper very carefully, I’m also including my more specific comments and critical remarks, divided into “Substantive” and “Technical”.

Substantive problems

The convoluted algorithm includes a series of corrections, one after another. At some point the reader is lost as to the purpose of all those corrections. Perhaps it would be helpful to add a graph showing the distribution of the sample 1000 structures?

Response:

Figure 6 and Figure 3 Panels D-F show the improvement provided with each refinement. By the final refinement, each atom type density ratio distribution are much more consistent. We have now quantified this improvement in terms of a shrinking standard deviation directly indicated in Figure 6. In the main text, we highlight this improvement as follows:

“Moreover, the standard deviation of the overall distribution drops to 0.09, representing a 40% decrease in the calculated density ratio variability.”

Issue 2:

The Authors should calibrate their method with ultrahigh-resolution PDB crystal structures which provide accurate estimate of electron density levels in e/A3 because they have a large Ewald sphere of very accurate F(hkl) data and in addition allow reliable estimation of F(000) since the (nearly) complete atomic content of the unit cell is practically known. The recommended examples would be the PDB entries 3NIR (0.48 A, crambin, highest resolution but unsatisfactory refinement), 1EJG (0.54 A, crambin) (comparison of the two crambin structures would provide an interesting “internal standard”) and 2VB1 (0.65 A, lysozyme) for proteins, and 3P4J (0.55 A, Z-DNA) for nucleic acids.

I also note that, infrequent as they are, occasionally ultrahigh-resolution macromolecular structures are presented with the electron density maps expressed on the absolute scale (e.g. Addlagatta et al. (2001) Acta Cryst. D57: 649-663). Such maps could also be used to validate the scaling procedures proposed in this work.

Response:

As Figure S1 shows, calibrating to the ultrahigh-resolution alone will produce worse results for the majority of the wwPDB entries which consist of lower resolution entries. In fact, the final radii appear optimized to the median wwPDB resolution:

“This figure also implies that the current radii are likely optimized for the median resolution of x-ray crystallographic structures present in the wwPDB.”

Issue 3:

p3, “These low-quality regions arise from structural model and electron density mismatches…”, actually, most often the low-quality regions arise because of absence of electron density.

Response:

The lack of electron density is one implied outcome of regional protein mobility; however, based on the reviewer’s comments we have made this point explicit: 

“These low-quality regions arise from structural model and electron density mismatches that can be due to a variety of reasons including problems with regional protein mobility that can often lead to an apparent lack of electron density [8-10], data processing [11, 12], or model fitting [10, 13, 14].”

Issue 4:

p6, I am surprised that the Authors have completely overlooked the paper by Tickle (2012) Acta Cryst. D68: 454-467, which is the standard classical reference for sigma-contoured electron density maps and more. Also, there is a good discussion there about the radii of atoms that cover 95% of electron density, and about the influence of B-factors and resolution (see 5.6. The limiting radius of the atomic density; as well as Table 3 and Fig. 11 therein).

Response:

We thank the reviewer for highlighting this important reference to add! We know this paper, but had not recognized the its significance with respect to the relationships between atomic radii, B-factor, and resolution. We have added it in the following context in the Discussion section: 

“These relationships between atomic radii, B-factors, and resolution with respect to observed electron density have been described previously [32]; however, Figures 5 and S1 provide a useful visualization of these relationships.” 

Issue 5:

Fig. 2 caption; it is correct to include H atoms in the electronic inventory of their “carriers” if H atoms were not included in Fc calculations. However, as is very often the case, if H atoms were included in Fc, then this strategy is incorrect. Moreover, if the electron density of H atoms is added to the bound atom, then interpretation by a simple spherical volume around that atom will lead to systematic errors. In addition, the resolution of the electron density map is also important.

Response:

We immensely thank the reviewer for this bit of information!! In the future, we can optimize two sets of radii based on hydrogen electrons included or not. We should be able to reliably predict if the Fc is calculated with or without the hydrogens. But this would be in a future improvement of the methodology. Given that all of the electron density maps we analyzed are generated by the PDB in Europe, we would assume that they are consistently generated with either Fc calculations including or excluding hydrogen electrons. Based on this consistency, the radii will be optimized to compensate for either situation.

Issue 6:

I don’t understand the idea of dividing the formula for F(000) in eq. (6) by V. Structure factors F (including F(000), of course), are expressed directly in electrons (e). Also, counting the electron contribution of the solvent molecules is necessary, but in most cases we miss a lot of water molecules (not included in the model). At low resolution, we cannot count the water molecules at all. What could be done, however, would be to estimate the number of water molecules from Matthews volume and specific density of water (1 g/cm3) and add their electrons to F(000).

Response:

Again, we thank the reviewer for their comments! We tried many permutations of calculating F000 based on the representations we had seen in different sources. Based on this information, we may try additional approaches in the future. However, given that there are different methods for deriving F000 for structures of different resolutions, our approach does provide one general approach for deriving a conversion factor that is somewhat resolution agnostic. We have made this point indirectly in the manuscript:

“Furthermore, the bulk solvent is estimated in different ways depending on resolution [31, 32].”

Issue 6:

I wonder, how much the re-scaled difference maps show-cased on p15 and Fig. 10 would differ from normal mFo-DFc and 2mFo-DFc maps. I think such a comparison should be illustrated.

Response:

We did not rescale the maps. We just labeled the discrepant density blobs in terms of eletrons of discrepancy. However, we did add the following to the Fgure 10 legend:

“The blue lattice represents the significant density regions in the 2Fo-Fc map, while the green lattice represents the significant positive discrepant density blobs and the red lattice represents the significant negative discrepant density blobs, both from the Fo-Fc map.”

Issue 7:

The method should be extended (in the future?) to explicitly apply to nucleic acids as well. Nucleic acids constitute an important segment of PDB structures.

Response:

This is our future plan and the software is capable of generating the radii for other biomacromolecules.

Issue 8:

To be of general applicability, a method like this should produce electron density maps in mtz or other ccp4/coot/pymol-readable format, so that they could be easily loaded and displayed for visualization.

Response:

The pdb-eda software analyzes electron density maps, it does not generate them. In particular, it is designed to download and analyze the electron density maps provided by the PDB in Europe REST interface.

Issue 9:

Technical remarks

p3, “activities happens”, singl./pl. problem.

Response:

Thanks! Fixed.

Issue 10:

p6 and elsewhere, the Greek letter sigma (and other symbols) is misprinted as a funny character.

Response:

Thanks for catching this. Looks like some of our Greek symbols got converted to weird symbols going back and forth between Windows and Mac versions of Word. Fixed.

Issue 10:

I couldn’t find anywhere in the ms the key information about the density of the grid over which the electron density is calculated and analyzed.

Response:

This manuscript is about deriving the electron density conversion factor. We have another publication where we apply methods that sum up absolute electron density discrepancy in a given region of a wwPDB entry: Sen Yao and Hunter N.B. Moseley. "Finding high-quality metal ion-centric regions across the worldwide Protein Data Bank" Molecules 24, 3179 (2019).

Issue 11:

Fig. 2 caption, “20 common amino acid”, singl./pl. problem.

Response:

Thanks again! Fixed.

Issue 12:

Correct the grammar in Fig. 2 (“by atom the b-factor”).

Response:

We fixed this as follows: “Adjust electron density ratios using atom-specific B-factor correction”

Issue 13:

p7, “total number of electrons are”, singl./pl. problem.

Response:

Thanks! Fixed.

Issue 14:

On p8, the term “chain deviation fraction” is suddenly changed to “chain fraction” (without definition).

Response:

Nice catch. Fixed.

Issue 15:

p8 and elsewhere, “on 100 random structures”, change to “of…”.

Response:

Respectfully, we believe this grammar is correct.

Issue 16:

p10, “Fig. 3. Sina plots of density ratio for atoms, residues, and chains”, perhaps it would be helpful to add a short description a sina plots, for example as in https://ggforce.data-imaginist.com/reference/geom_sina.html: “Sina plot is an enhanced jitter strip chart, where the width of the jitter is controlled by the density distribution of the data within each class.”.

Response:

We described them as follows, which includes a reference:

“If we simply use the radii from the literature and do not apply any correction, the median of atom density ratio shows that the density ratios are inconsistent within a single structure for atoms, residues, and chains, as illustrated in Fig 3 Panels A-C as Sina plots, an enhanced jitter strip chart that visualizes the distribution of each dataset for better visual comparison [30]”

Issue 16:

Fig. 3, frankly, I don’t see much difference between the top and bottom panels… Moreover, it seems to me that the x axis of panels C and F shows residues, not chain IDs; I am confused…

Response:

There is quite a bit of difference at the residue level (comparing panels A and D) before and after radii optimization. There is less difference at the residue and chain level, but these are included to highlight how stable the density ratio is at the chain level versus the residue and atom level. We highlight these points in the main text:

“If we simply use the radii from the literature and do not apply any correction, the median of atom density ratio shows that the density ratios are inconsistent within a single structure for atoms, residues, and chains, as illustrated in Fig 3 Panels A-C as Sina plots, an enhanced jitter strip chart that visualizes the distribution of each dataset for better visual comparison [30]. The atom density ratios span over the largest range, while the chain density ratio has the smallest range.”

Issue 17:

The “optimized” atomic radii are usually >= the “original” radii. An outstanding exception in Table 1 is S. Any reason why?

Response:

We suspect that there is some old assumption made a long time ago that is either baked into most x-ray crystallographic software or into the original sulfur radius estimation. What it is exactly, we do not know, but we would wager that it is some assumption about sulfur behaving like another element when in reality it does not. However, we are not willing to make this type of statement in this manuscript. We have already highlighted this point about the sulfur as follows:

“The radius of sulfur changes the most as compared to the other elements. This could be due to how most software construct electron density data from structure factor and associated phase via Fourier transforms. The electron density is approximated with a Gaussian distribution, and its variance is affected mainly by the B-factor. Thus, the final optimized radius is a combination of the actual radius of the atom, the displacement of an atom center, as well as the thermal motion of the atom (B-factor).”

Issue 18:

p13, last sentence before new section, don’t begin sentence with “And…”.

Response:

Technically, this is not considered bad writing form anymore, but that is a matter of perspective. However, we went ahead and changed it to:

“Moreover, studying the behavior of sulfur atoms could be useful for other less common elements such as metal ions.”

Issue 19:

p13, explain “distribution modes”.

Response:

The phrase is “tighter distributions with modes above 0”. For a unimodal distribution, the distribution has one mode. In this phrase, we are talking about the mode of each chain deviation fraction distribution associated with each atom type for high-resolution structures (< 1.5 angstroms). All of these distributions have a mode above zero as illustrated in Figure S1.

Issue 20:

p13, “It was then applied to the all PDB structures that has usable electron density data”, several grammatical errors in this sentence.

Response:

Thanks! All grammatical errors are fixed as follows:

“It was then applied to all PDB structures that had usable electron density data, and the results are shown in Fig 7.”

Issue 21:

Fig. 8 caption talks about 2mFo-DFc maps, but panels A/B are supposed to correspond to 2Fo Fc/Fo Fc maps.

Response:

Our mistake. We have change the Figure 8 title to: “Histogram of the mean value of electron density maps provided by PDBe.”

Issue 22:

Something wrong with the first sentence of Fig. 9 caption.

Response:

We change the Figure 9 title to: “The absolute scale of the density ratio for all structures in the PDB.”

Issue 23:

p14, “with a local region.”, perhaps “within a local region.”.

Response:

Thanks. Fixed.

Issue 24:

p15, difference electron density of “16 and 29 e” sounds like serious overshooting. Should be metal or halide ions, or at least S.

Response:

Our point exactly. The number of electrons of discrepancy can inform the likely issue in the structural model.

Issue 25:

Fig. S1, the axes need proper labels and units. The values of the ticks on the x axes are squeezed too much and unreadable. The plots, with the current caption, are not very useful.

Response:

We have fixed the overlap in the values. However, the x-axis deals with a fraction that is unitless.

Issue 26:

Tables S1 and S2 with the inventory of bonds and atoms in standard amino acid residues are banal and could be omitted.

Response:

This is supplementary material. It does not matter if it is banal. We try to always error on the side of completeness with supplemental material.

Issue 27:

Please note that the proper symbol to be used for Atomic Displacement Parameter (temperature factor) is B-factor.

Response:

We have fix this across the manuscript.

Issue 28:

I am glad to note that the GITHUB documentation and examples seem to work well. I note that GITHUB reports version 1.0.1. and PythonPackageIndex version 1.1.0. There are also some shortcomings and bugs. For example, the command:

python3 -m pdb_eda single 3han 3han.all.csv --all --out-format=csv

instead of human-readable printouts, returns a printout of rather useless python objects.

Execution of:

python3 -m pdb_eda single 3han 3han.all.csv --all --out-format=csv

generates an error message:

AttributeError: 'int' object has no attribute 'aggregateCloud'

which is not very helpful.

Response:

We do try very hard to develop well-documented scientific codebases. We also beg to differ about Python objects being useless, but we will work on developing more human-readable printouts in future releases. Also, we have limited control over the quality of the error messages that the Python interpreter generates.

---

## [Decision Letter · Decision Letter 1]

7 Jul 2020

PONE-D-20-10145R1

A chemical interpretation of protein electron density maps in the worldwide protein data bank

PLOS ONE

Dear Dr. Moseley,

Thank you for submitting your manuscript to PLOS ONE. After careful consideration, we feel that it has merit but does not fully meet PLOS ONE’s publication criteria as it currently stands. Therefore, we invite you to submit a revised version of the manuscript that addresses the points raised during the review process.

Specifically, there are a few technical issues pointed out by the reviewer that need to be addressed.

We look forward to receiving your revised manuscript.

Kind regards,

Oscar Millet

Academic Editor

PLOS ONE

Reviewers' comments:

Reviewer's Responses to Questions

**Comments to the Author**

1. If the authors have adequately addressed your comments raised in a previous round of review and you feel that this manuscript is now acceptable for publication, you may indicate that here to bypass the “Comments to the Author” section, enter your conflict of interest statement in the “Confidential to Editor” section, and submit your "Accept" recommendation.

Reviewer #3: (No Response)

2. Is the manuscript technically sound, and do the data support the conclusions?

Reviewer #3: Yes

3. Has the statistical analysis been performed appropriately and rigorously? 

Reviewer #3: N/A

4. Have the authors made all data underlying the findings in their manuscript fully available?

Reviewer #3: Yes

5. Is the manuscript presented in an intelligible fashion and written in standard English?

Reviewer #3: Yes

6. Review Comments to the Author

Reviewer #3: (No Response)

7. PLOS authors have the option to publish the peer review history of their article (what does this mean?). If published, this will include your full peer review and any attached files.

Reviewer #3: No

---

## [Author Response · Author response to Decision Letter 1]

9 Jul 2020

Reviewer 3:

In view of the rebuttal, in particular emphasising that the audience of this paper are not professional crystallographers but less trained consumers of the PDB, as well as highlighting that the application of the proposed method and software lies in PDB-wide analyses, I reluctantly tend to agree that this manuscript could be accepted after additional minor revision. I am also amused by the correct unmasking of Referee 3 (i.e. myself) as a (well trained, I supposed) crystallographer! However, I have the impression that in many cases the Authors have glossed over the highlighted problems too easily, sometimes actually evading to address the issue at hand directly. 

Response:

We thank the reviewer for seeing the utility of our software for the general consumer of the PDB. We were not trying to gloss over issues raised by the reviewers, but there were quite a few issues raised by all three reviewers, some issues with more merit than others. The revisions and response document take real effort to generate.

Issue 1:

One such remark is “my” Issue 6 (first instance with this issue number, as the Authors’ numbering of issues has errors), in which I pointed out that F(000) (as any other structure factor) is expressed in electrons. Therefore, the division of formula (6) by unit cell volume (V) is incorrect. 

Response:

We misunderstood the point being made here the first time, because we were focused on how to use F(000) for correction. Thank you for raising it again! We would not want this kind of definition error in the published literature. We have made the following revisions:

“Unfortunately, not all structure factor programs provide the F000 value. So as an estimation, we add up the numbers of electrons for all the atoms of a model in the unit cell, including symmetry structure units and modeled water molecules, 

F000 = ∑n_t Z_t (6)

Where nt is number of atoms of element t in the asymmetric unit and Zt is the number of electrons (atomic number) of element t. This estimated F000 term is then divided by the unit cell volume V in Å3 and added to the density values of all the voxels.

”

Issue 2:

Also, I think that the Authors are too quick to dismiss bug reports and to defer action to the future. Reports like the one below are useless for untrained users and can be certainly handled in a better way by error-handling routines included in the software package.

>> python3 -m pdb_eda single 6xyz 6xyz.atom.csv --atom --out-format=csv

>> AttributeError: 'int' object has no attribute 'aggregateCloud'

Response:

It is not that we are dismissing a bug report. We handle all issues posted on our GitHub repositories. Please look at https://github.com/MoseleyBioinformaticsLab to see where we address issues raised by users (for example, https://github.com/MoseleyBioinformaticsLab/GOcats/issues). However, we cannot anticipate all possible errors raised by the Python interpreter and provide a meaningful response to the user. All we can reasonably do is fix bugs and return more meaningful error messages for the bugs or edge cases that we find ourselves or are pointed out to us by others. Towards this end, we have verified the error and concluded that it is due to a flaw in the command line interface where object creation was not correctly tested for success. We have created an issue on our GitHub pdb-eda repository based on what the reviewer has provided and our testing of the error: https://github.com/MoseleyBioinformaticsLab/pdb_eda/issues/1 . We expect to have this issue fixed when we roll out a larger set of updates to the codebase in the next 2-4 weeks. Please be patient with us.

---

## [Editor Report · Decision Letter 2]

16 Jul 2020

A chemical interpretation of protein electron density maps in the worldwide protein data bank

PONE-D-20-10145R2

Dear Dr. Moseley,

We’re pleased to inform you that your manuscript has been judged scientifically suitable for publication and will be formally accepted for publication once it meets all outstanding technical requirements.

Kind regards,

Oscar Millet

Academic Editor

PLOS ONE
---

## [Editor Report · Acceptance letter]

20 Jul 2020

PONE-D-20-10145R2 

A chemical interpretation of protein electron density maps in the worldwide protein data bank 

Dear Dr. Moseley:

I'm pleased to inform you that your manuscript has been deemed suitable for publication in PLOS ONE. Congratulations! Your manuscript is now with our production department. 

Kind regards, 

on behalf of

Dr. Oscar Millet 

Academic Editor

PLOS ONE